# A Generalizable Physics-Enhanced State Space Model for Long-Term Dynamics Forecasting in Complex Environments

Yuchen Wang [1]   Hongjue Zhao [2]   Haohong Lin [3]   Enze Xu [1]   Lifang He [4]   Huajie Shao [1]

## Abstract

This work aims to address the problem of long-term dynamic forecasting in complex environments where data are noisy and irregularly sampled. While recent studies have introduced some methods to improve prediction performance, these approaches still face a significant challenge in handling long-term extrapolation tasks under such complex scenarios. To overcome this challenge, we propose Phy-SSM, a generalizable method that integrates partial physics knowledge into state space models (SSMs) for long-term dynamics forecasting in complex environments. Our motivation is that SSMs can effectively capture long-range dependencies in sequential data and model continuous dynamical systems, while the incorporation of physics knowledge improves generalization ability. The key challenge lies in how to seamlessly incorporate partially known physics into SSMs. To achieve this, we decompose partially known system dynamics into known and unknown state matrices, which are integrated into a Phy-SSM unit. To further enhance long-term prediction performance, we introduce a physics state regularization term to make the estimated latent states align with system dynamics. Besides, we theoretically analyze the uniqueness of the solutions for our method. Extensive experiments on three real-world applications, including vehicle motion prediction, drone state prediction, and COVID-19 epidemiology forecasting, demonstrate the superior performance of Phy-SSM over the baselines in both long-term interpolation and extrapolation tasks. The code is available at https://github.com/511205787/Phy_SSM-ICML2025.

## 1. Introduction

Dynamical systems have been widely applied across a broad range of real-world domains, including autonomous driving (Kong et al., 2015; Rajamani, 2011), epidemiology (Nicho, 2010), and climate science (Ionides et al., 2006).

Generally, dynamical systems are often governed by underlying physical laws. Motivated by this, some studies have developed physics-enhanced machine learning models (O'Driscoll et al., 2019; Cicirello, 2024) for to enhance the generalization ability of dynamics forecasting by incorporating known physical laws, such as energy conservation and differential equations (Greydanus et al., 2019; Raissi et al., 2019). A common assumption in these methods is that the physical laws governing system dynamics are *fully known* as prior knowledge. However, in practice, it is challenging to obtain the complete governing equations for complex dynamical systems using first principles (Linial et al., 2021; Gentine et al., 2018). On the other hand, dynamical systems like autonomous vehicles operate in complex and unknown environments such as inclement weather conditions, their sensors often suffer from faults or mismatched clocks (Dabrowski & Rahman, 2019; Huang et al., 2023), resulting in noisy and irregularly sampled data.

To address these issues, few recent works (Linial et al., 2021; Takeishi & Kalousis, 2021; Yang et al., 2022) have developed partially known physics-enhanced machine learning models for noisy, regular data. While these methods perform well in interpolation tasks, they often struggle with long-term extrapolation tasks with *irregular* data. An illustrative example is provided in Appendix A. This limitation arises from their solutions heavily relying on initial conditions, lacking an effective mechanism to dynamically refine predictions based on subsequent observations (Chen et al., 2024; Kidger et al., 2020). Thus, the question is: how to enhance accuracy and generalization for long-term dynamics forecasting with *noisy, irregular data*?

In this work, we propose Phy-SSM, a generalizable method that incorporates partially known physics into deep state-space models (SSMs), as shown in Fig. 1. Our *motivation* is that deep SSMs (Gu et al., 2021; Smith et al.; Gu & Dao, 2023) can not only effectively capture long-range dependen-

[1]William & Mary [2]University of Illinois at Urbana-Champaign [3]Carnegie Mellon University [4]Lehigh University. Correspondence to: Huajie Shao <hshao@wm.edu>.

*Proceedings of the 42$^{st}$ International Conference on Machine Learning*, Vancouver, Canada. PMLR 267, 2025. Copyright 2025 by the author(s).

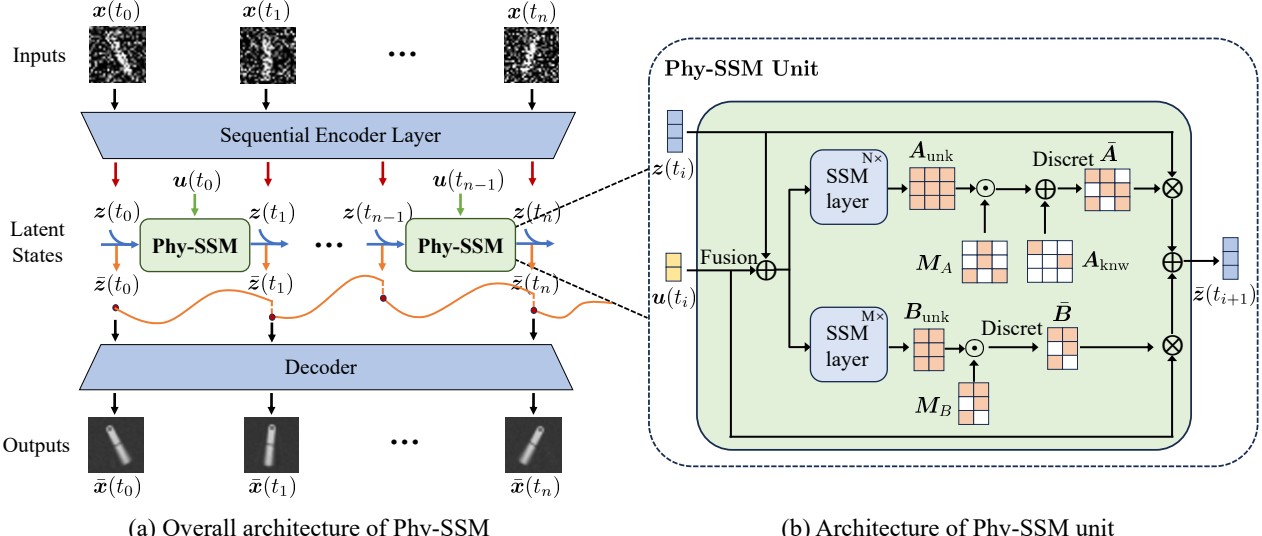

(a) Overall architecture of Phy-SSM

(b) Architecture of Phy-SSM unit

*Figure 1.* (a) The overall architecture of Phy-SSM, consisting of three components: a sequential encoder, Phy-SSM Unit, and a decoder. (b) Detailed architecture of Phy-SSM unit.

cies in sequential data but also model continuous dynamical systems. However, developing Phy-SSM involves addressing two main *challenges*: (i) seamlessly integrating partial physics knowledge into the model architecture, and (ii) facilitating long-term predictions in the presence of noisy and irregularly sampled data. To tackle the first challenge, we develop a novel Phy-SSM unit that decomposes partially known dynamics into known and unknown state matrices, as shown in Fig. 1(b). This decomposition represents a significant advancement compared to existing deep SSMs (Gu & Dao, 2023; Gu et al., 2022). For the second challenge, we introduce a physics state regularization term to further constrain the estimated latent states by the encoder to comply with the system dynamics. Furthermore, we provide a theoretical analysis of the uniqueness of solutions for modeling partially known dynamical systems. Finally, we evaluate the proposed Phy-SSM on three real-world applications: vehicle motion prediction, drone state prediction, and COVID-19 epidemiology forecasting. The results demonstrate that our method significantly outperforms baseline methods for long-term interpolation and extrapolation tasks. These findings highlight the effectiveness of incorporating partial physical knowledge into deep SSMs for improving predictive generalization in complex, real-world scenarios.

**Our contributions are four-fold**: 1) We propose Phy-SSM, a novel approach that integrates partially known physics into state-space models to improve generalization for long-term forecasting in complex environments; 2) To enhance long-term prediction accuracy, we introduce a physics state regularization term that constrains latent states to align with system dynamics; 3) We offer a theoretical analysis of uniqueness of solutions for our method; and 4) Extensive experiments on three real-world applications demonstrate

that Phy-SSM significantly outperforms baseline methods in long-term dynamics forecasting.

## 2. Related Work

**Physics-Enhanced Machine Learning (PEML)**. Depending on how underlying physics knowledge is incorporated into models, PEML (Faroughi et al., 2022; O'Driscoll et al., 2019; Cicirello, 2024) can be classified into two main types:

(i) *Physics-Informed Loss Function*. This method incorporates physical laws into the loss function as a soft constraint, ensuring that ML models remain consistent with the laws of physics (Baydin et al., 2018; Chen et al., 2020; Wang et al., 2023; Raissi et al., 2020; Yu et al., 2022; Raissi et al., 2019). A typical line of this method is PINN (Lu et al., 2021; Raissi et al., 2019) that integrates differential equations into the loss. However, such methods often struggle to extrapolate beyond the training distribution (Bonfanti et al., 2024; Kim et al., 2021), since they are trained to conform to the solutions within a pre-specified domain.

(ii) *Physics-Informed Architecture Design*. This approach tries to embed physical principles into the design of ML architectures to enhance model generalization. Some works focused on neural ordinary differential equations (NODEs) (Chen et al., 2018) that adopted deep neural networks (DNNs) to parameterize underlying ODEs. For instance, a recent study developed ContiFormer (Chen et al., 2024) that combines Transformer with NODEs to model continuous-time dynamics on irregular time series. This method, however, is very computationally expensive and may struggle to extract generalized physical representations for long-term predictions. Other studies designed

ML to model energy-conserving systems by complying with Hamiltonian mechanics (Greydanus et al., 2019; Bacsa et al., 2023; Zhong et al.) or Lagrangian mechanics (Cranmer et al., 2020; Lutter et al., 2018). While these methods improve model generalization ability due to their embedded physics inductive bias, their reliance on specific physics knowledge limits their broader applicability as a general-purpose approach.

In summary, these existing methods did not consider real-world dynamical systems where obtaining complete physics knowledge is often infeasible (Linial et al., 2021).

**Partially Known Physics-Based ML**. Some works have focused on partially known physics-based machine learning (ML). For instance, SINDy Autoencoders (Champion et al., 2019) identified physical laws directly from data by optimizing a linear combination of predefined functions. Phy-Taylor (Mao et al., 2023) integrated system dynamics into its architecture design. However, these approaches rely on finite difference techniques to estimate derivatives, making them only applicable to noise-free and regular data (Champion et al., 2019; Yin et al., 2021).

Recently, few studies (Yin et al., 2021; Linial et al., 2021; Wehenkel et al., 2023; Takeishi & Kalousis, 2021) have explored partially known physics-enhanced machine learning that use physics-based NODEs to model underlying dynamics. While these methods have proven effective for continuous-time modeling, their reliance on NODEs presents challenges in accurately capturing nonlinear and time-variant systems over long time horizons. This limitation arises from NODEs' heavy reliance on initial conditions, which hinders their ability to capture subsequent long-term sequence correlations. Additionally, these methods are only employed to regular data rather than irregular data in the experiments.

Different from prior work, we propose a generalizable model that leverages partial physics knowledge to enhance the generalization of long-term dynamics forecasting with noisy, irregular data.

# 3. Preliminaries

**Notations**. The detailed descriptions of important notations are presented in Table 6 in Appendix B.

## 3.1. Problem Statement

We first review the background of dynamical systems and then describe our research problem.

**Background.** Given a set of observations $\boldsymbol{x}(t)$ from a dynamic system. We assume that the observed trajectory is governed by the underlying system dynamics $\boldsymbol{z}$ through a fixed emission function $\boldsymbol{g}$:

$$\boldsymbol{x}(t) = \boldsymbol{g}\left(\boldsymbol{z}(t)\right), \tag{1}$$

where $\boldsymbol{x} : [0, T] \to \mathcal{X} \subseteq \mathbb{R}^{d_x}$ and $\boldsymbol{z} : [0, T] \to \mathcal{Z} \subseteq \mathbb{R}^{d_z}$ represent the observations and full system states, respectively. In practice, the system states $\boldsymbol{z}(t)$ are hardly observable, governed by the underlying system dynamics below:

$$\frac{\mathrm{d}\boldsymbol{z}(t)}{\mathrm{d}t} = \boldsymbol{f}\left(\boldsymbol{z}(t), \boldsymbol{u}(t)\right), \tag{2}$$

where $\boldsymbol{u}(t)$ represents the control input that influences the system, and $\boldsymbol{f}$ denotes a governing equation of system dynamics, which is often partially known.

**Our Problem.** *The goal of this work* is to predict the long-term trajectories of a dynamical system in complex environments where data are noisy and irregularly sampled. Specifically, the inputs include the observational sequence $[\boldsymbol{x}(t_0), \boldsymbol{x}(t_1), \cdots, \boldsymbol{x}(t_n)]$ with $n + 1$ data points, where $0 = t_0 < t_1 < \cdots < t_n = T$, and the control input sequence $[\boldsymbol{u}(t_0), \cdots, \boldsymbol{u}(t_n), \cdots, \boldsymbol{u}(t_{n+l})]$, where $l$ is the length of the prediction window.

Our tasks will involve: i) *Interpolation*: Predict the sequence $[\bar{\boldsymbol{x}}(t_0), \bar{\boldsymbol{x}}(t_1), \cdots, \bar{\boldsymbol{x}}(t_n)]$ using the learned dynamics. Here, $\bar{\boldsymbol{x}}(t_i)$ is the predicted observation at $t_i$, obtained using our method. ii) *Extrapolation*: Forecast the future trajectories $\bar{\boldsymbol{x}}(t)$ for $t \in (t_n, t_{n+l}]$.

## 3.2. Structured State Space Model

We introduce the foundation of Structured State Space Models (S4) (Gu et al., 2021), which serves as the core component of the proposed Phy-SSM. S4 is an emerging neural architecture built on classical state space models (SSMs) from control theory, designed to capture long-range dependencies in sequential data. To better understand S4, we first review the basic knowledge of classical SSMs as follows:

$$\begin{aligned}
\dot{\boldsymbol{z}}(t) &= \boldsymbol{A}\boldsymbol{z}(t) + \boldsymbol{B}\boldsymbol{u}(t), \\
\boldsymbol{y}(t) &= \boldsymbol{C}\boldsymbol{z}(t) + \boldsymbol{D}\boldsymbol{u}(t),
\end{aligned} \tag{3}$$

where $\boldsymbol{A} \in \mathbb{R}^{n \times n}$, $\boldsymbol{B} \in \mathbb{R}^{n \times m}$, $\boldsymbol{C} \in \mathbb{R}^{p \times n}$, and $\boldsymbol{D} \in \mathbb{R}^{p \times m}$ are the state, input, output, and feedthrough matrices, respectively. In addition, $\boldsymbol{u}(t) \in \mathbb{R}^m$ represents the input signal, $\boldsymbol{z}(t) \in \mathbb{R}^n$ denotes the state variables, and $\boldsymbol{y}(t) \in \mathbb{R}^p$ represents the outputs. Note that $\boldsymbol{D}\boldsymbol{u}(t) = \boldsymbol{0}$, as $\boldsymbol{D}\boldsymbol{u}(t)$ can be interpreted as a skip connection.

For practical applications involving discrete sequences, the continuous model needs to be discretized. Common discretization methods, such as bilinear transformations (Tustin, 1947), can be employed. By discretizing with a step size $\Delta$, the system is expressed by the following linear recurrence relations:

$$\begin{aligned}
\boldsymbol{z}_n &= \bar{\boldsymbol{A}}\boldsymbol{z}_{n-1} + \bar{\boldsymbol{B}}\boldsymbol{u}_n, \\
\boldsymbol{y}_n &= \bar{\boldsymbol{C}}\boldsymbol{z}_n,
\end{aligned} \tag{4}$$

where $\bar{A}$, $\bar{B}$, and $\bar{C}$ are the discrete-time parameters derived from $A$, $B$, and $C$ with the step size $\Delta$. These parameters preserve the dynamics of the continuous model in the discretized setting, enabling effective modeling of sequential data. S4 (Gu et al., 2021) extends the use of SSMs for modeling long sequences. In particular, S4 leverages a specialized matrix initialization technique called HiPPO (Gu et al., 2020) to efficiently maintain information from past inputs. Building on S4, multiple variants of deep SSMs such as S5 (Smith et al.) and Mamba (Gu & Dao, 2023) have been proposed in recent years.

Among these, recent work (Smith et al.) introduced a S5 model that employs parallel scans to accelerate training in a recurrent mode. This design allows the model to efficiently handle time-varying SSMs. Its key advantages include: (i) It can capture long-term data dependencies through the HiPPO memory, and (ii) Its continuous-time formulation enables the effective modeling of irregularly sampled data. Inspired by these, we developed a physics-enhanced SSM that incorporates partial physics knowledge for long-term dynamics forecasting in complex environments.

## 4. Proposed Method

### 4.1. Framework of Phy-SSM

As mentioned in Sec. 3.1, our goal is to enhance long-term predictions of dynamical systems when data are noisy and irregularly sampled. To achieve this, we propose a novel Phy-SSM that incorporates partially known physics knowledge into the model design, as shown in Fig. 1. Phy-SSM is composed of three main components: a sequential encoder layer, the Phy-SSM Unit, and a decoder, with the Phy-SSM Unit serving as the core. The key idea behind Phy-SSM is to first employ a sequential encoder to encode the inputs to approximate the posterior distribution of the latent system state. Next, the Phy-SSM Unit jointly takes in the past latent state and control inputs to predict the next latent states. Finally, the latent states generated by the Phy-SSM Unit are fed into the decoder to produce the final output. Below, we elaborate on each component in the proposed framework.

**Encoder for Posterior Probability Estimation**. We first adopt a sequential encoder $\varphi$ to estimate the posterior distribution of the latent system states $z(t)$ based on observations $x(t)$. The sequential encoder is a simplified structured SSM that can handle irregular data while introducing a memory variable $h(t)$ to capture long sequence correlations. Specifically, the approximate posterior $z(t_i)$ at each time step depends not only on $x(t_i)$ but also on the memory from the previous time step, $h(t_{i-1})$. Mathematically, we have

$$z(t_i) \mid x(t_i) \sim \mathcal{N}\left(\hat{\boldsymbol{\mu}}_z(t_i), \operatorname{diag}(\hat{\boldsymbol{\sigma}}_z^2(t_i))\right), \quad (5)$$

where

$$\hat{\boldsymbol{\mu}}_z(t_i), \hat{\boldsymbol{\sigma}}_z(t_i) = \varphi\left(x(t_i), h(t_{i-1})\right)$$

and $\hat{\boldsymbol{\mu}}_z(t)$, $\hat{\boldsymbol{\sigma}}_z^2(t)$ respectively represent the mean and variance of the learned posterior distribution. The process is stochastically approximated using the reparameterization trick (Kingma, 2013).

Next, we need to approximate the posterior distribution $q(z(t_{\leq n}) \mid x(t_{\leq n}))$ based on the above Eq. (5). To reduce computational costs, inspired by prior work (Girin et al., 2020), the posterior distribution can be simplified as

$$q(z(t_{\leq n}) \mid x(t_{\leq n})) \approx \prod_{i=0}^{n} q(z_i \mid x(t_{\leq i})). \quad (6)$$

**Phy-SSM Unit.** The primary role of the Phy-SSM Unit is to enforce known physical laws while simultaneously learning unknown dynamics from sequential data. Assuming that the latent system states conform to certain physical dynamics, the Phy-SSM Unit leverages the latent states and control input from the previous time step as input for generating physics-consistent predictions. However, developing the Phy-SSM unit poses two key challenges: *1) incorporating partial physics knowledge into model design*, and *2) accurately modeling the unknown dynamics for long-term predictions*.

To address the *first* challenge, we enforce known physics knowledge through dynamics decomposition. Specifically, consider the system dynamics in Eq. (2), we decompose it into known and unknown parts as follows:

$$\begin{aligned}
\frac{\mathrm{d}z(t)}{\mathrm{d}t} &= \boldsymbol{f}\left(z(t), \boldsymbol{u}(t)\right) \\
&= \boldsymbol{f}_{\mathrm{knw}}\left(z(t), \boldsymbol{u}(t)\right) + \boldsymbol{f}_{\mathrm{unk}}\left(z(t), \boldsymbol{u}(t)\right).
\end{aligned} \quad (7)$$

Inspired by prior work (Brunton et al., 2016), we transform the system dynamics into a linear SSM by extending the state $z$ to a new state $\bar{z}$, which includes $z$ and can additionally incorporate *nonlinear terms or constants*. This method can help to *represent nonlinear systems in a linear manner*. By doing this, it enables us to: (i) use matrix calculation to improve computing efficiency, and (ii) embed knowledge into matrices directly.

Based on this motivation, we convert the above Eq. (7) into the following linear SSM formula.

$$\begin{aligned}
\frac{\mathrm{d}\bar{z}(t)}{\mathrm{d}t} &= \boldsymbol{A}(t)\bar{z}(t) + \boldsymbol{B}(t)\boldsymbol{u}(t) \\
&= (\boldsymbol{A}_{\mathrm{knw}}(t) + \boldsymbol{A}_{\mathrm{unk}}(t))\bar{z}(t) + \boldsymbol{B}_{\mathrm{unk}}(t)\boldsymbol{u}(t),
\end{aligned} \quad (8)$$

where $\boldsymbol{A}$ is the state matrix, and $\boldsymbol{B}$ is the input matrix. The extended state is defined as $\bar{z} = [z^{\top}, \boldsymbol{\psi}(z)^{\top}]^{\top} \in \mathbb{R}^{d_{\bar{z}}}$, where $\boldsymbol{\psi}(z) \in \mathbb{R}^{d_{\psi}}$ represents additional extended terms. Here, $\boldsymbol{A}_{\mathrm{knw}} \in \mathbb{R}^{d_{\bar{z}} \times d_{\bar{z}}}$, $\boldsymbol{A}_{\mathrm{unk}} \in \mathbb{R}^{d_{\bar{z}} \times d_{\bar{z}}}$, and $\boldsymbol{B}_{\mathrm{unk}} \in \mathbb{R}^{d_{\bar{z}} \times d_u}$. Specifically, $\boldsymbol{A}_{\mathrm{knw}}(t)\bar{z}(t)$ denotes the known physical dynamics, while $\boldsymbol{A}_{\mathrm{unk}}(t)\bar{z}(t)$ represents the unknown system dynamics. $\boldsymbol{B}_{\mathrm{unk}}(t)\boldsymbol{u}(t)$ models the influence of control inputs. In Eq. (8), we omit $\boldsymbol{B}_{\mathrm{knw}}$ since the influence

of control inputs is often unknown; otherwise, it can be treated in the similar way as $\boldsymbol{A}_{\mathrm{knw}}$. After decomposition, it becomes straightforward to enforce explicit physical knowledge into the model. A detailed example of illustrating this transformation process is provided in Sec. 4.3.

To address the *second* challenge, inspired by the powerful HiPPO memory mechanism (Gu et al., 2020), we utilize multi-layer structured SSMs to model continuous unknown dynamics. The structured SSMs enable the Phy-SSM unit to memorize long-term historical patterns and accurately estimate unknown dynamics.

Based on the above ideas, we present the detailed structure of the Phy-SSM Unit, as illustrated in Fig. 1(b). Specifically, the Phy-SSM Unit involves the following three key steps for physics-based latent state prediction.

*1) Learning unknown continuous functions $\tilde{\boldsymbol{A}}_{\mathrm{unk}}(t)$ and $\tilde{\boldsymbol{B}}_{\mathrm{unk}}(t)$ using structured SSMs.* Multi-layer structured SSMs take $\boldsymbol{z}$ and $\boldsymbol{u}$ as input to approximate the unknown continuous functions $\boldsymbol{A}_{\mathrm{unk}}(t, \bar{\boldsymbol{z}}, \boldsymbol{u}; \boldsymbol{\theta}_A)$ and $\boldsymbol{B}_{\mathrm{unk}}(t, \bar{\boldsymbol{z}}, \boldsymbol{u}; \boldsymbol{\theta}_B)$. The outputs of the structured SSMs are passed through a fully connected layer and reshaped into $\mathbb{R}^{d_{\bar{z}} \times d_{\bar{z}}}$.

*2) Knowledge mask to encode physics as hard constraints.* To constrain the model to learn only unknown terms, we implement a simple yet effective knowledge mask mechanism. Specifically, we introduce a binary knowledge mask $\boldsymbol{M} \in \{0, 1\}^{d_{\bar{z}} \times d_{\bar{z}}}$ that is applied via the Hadamard product to the learned unknown terms. The positions with a value of 1 in the mask indicate that the corresponding dynamic item is permitted to be updated, while positions with a value of 0 block the influence of the item. The learned unknown dynamics are then refined as follows:

$$\begin{aligned} \boldsymbol{A}_{\mathrm{unk}}(t) &= \boldsymbol{M}_A \odot \tilde{\boldsymbol{A}}_{\mathrm{unk}}(t), \\ \boldsymbol{B}_{\mathrm{unk}}(t) &= \boldsymbol{M}_B \odot \tilde{\boldsymbol{B}}_{\mathrm{unk}}(t). \end{aligned} \quad (9)$$

For detailed knowledge mask design for different dynamical systems, we provide a guideline in Appendix I.

*3) Discretizing continuous dynamics to compute the next latent state.* After obtaining $\boldsymbol{A}_{\mathrm{knw}}(t)$ and $\boldsymbol{A}_{\mathrm{unk}}(t)$, we can compute the full system dynamics matrix $\boldsymbol{A}$ in Eq. (8). To discretize the continuous-time model for generating latent states, we use the bilinear method (Tustin, 1947), which converts the continuous state matrix $\boldsymbol{A}$ into its discrete approximation $\bar{\boldsymbol{A}}$. Notably, our model retains the continuous parameter $\boldsymbol{A}$, enabling it to handle irregularly sampled data. After discretization, the Phy-SSM unit generates the next time-step output:

$$\bar{z}(t_{i+1}) = \bar{\boldsymbol{A}}(t_i)\bar{z}(t_i) + \bar{\boldsymbol{B}}(t_i)\boldsymbol{u}(t_i). \quad (10)$$

**Prior Physics State Prediction for Decoder**. Unlike a standard VAE, where latent variables typically follow a

standard Gaussian distribution, we directly embed physics knowledge into a latent space. Consequently, the prior probability of the latent $\boldsymbol{z}(t)$ is defined as:

$$\boldsymbol{z}(t_i) \sim \mathcal{N}\left(\boldsymbol{\mu}_z(t_i), \mathrm{diag}(\boldsymbol{\sigma}_z^2(t_i))\right),$$

where $\boldsymbol{\mu}_z(t_i)$ and $\boldsymbol{\sigma}_z^2(t_i)$ represent the mean and variance of the physics-based prior distribution. To respect the physical dynamics, we use the Phy-SSM Unit to generate $\boldsymbol{\mu}_z(t_i)$ and $\boldsymbol{\sigma}_z^2(t_i)$. Specifically, the outputs of the Phy-SSM unit are passed through a linear map to compute $\boldsymbol{\mu}_z(t_i)$ and $\boldsymbol{\sigma}_z^2(t_i)$.

During the interpolation stage, the continuous Phy-SSM unit dynamically refines its predictions using the posterior from the preceding time step within the observation window. This approach effectively reduces prediction error stemming from potentially inaccurate initial conditions. During the extrapolation stage, Phy-SSM leverages accurate initial conditions and well-learned physics to perform predictions through autoregression. Once the full trajectory of physics latent states is obtained, according to (Girin et al., 2020), the decoder maps these latent states to the output, yielding

$$p(\boldsymbol{x}(t_{\leq n}), \boldsymbol{z}(t_{\leq n})) = \prod_{i=0}^{n} p(\boldsymbol{x}(t_i) \mid \boldsymbol{z}(t_i)) p(\boldsymbol{z}(t_i) \mid \boldsymbol{z}(t_{i-1})). \quad (11)$$

**Overall Objective.** The objective function comprises the the negative time step-wise variational lower bound (Chung et al., 2015) and the physics state regularization term below:

$$\mathcal{L} = \mathcal{L}_{\mathrm{VAE}} + \lambda \mathcal{L}_{\mathrm{reg}}, \quad (12)$$

where

$$\mathcal{L}_{\mathrm{VAE}} = -\sum_{i=0}^{n} \mathbb{E}_{q(\boldsymbol{z}(t_{\leq i}) \mid \boldsymbol{x}(t_{\leq i}))}[\mathcal{L}_{\mathrm{recon}}^{(i)} - \beta \mathcal{L}_{\mathrm{KL}}^{(i)}],$$

$$\mathcal{L}_{\mathrm{recon}}^{(i)} = \log p(\boldsymbol{x}(t_i) \mid \boldsymbol{z}(t_i)),$$

$$\mathcal{L}_{\mathrm{KL}}^{(i)} = \mathrm{KL}\left(q(\boldsymbol{z}(t_i) \mid \boldsymbol{x}(t_{\leq i})) \parallel p(\boldsymbol{z}(t_i) \mid \boldsymbol{z}(t_{i-1}))\right),$$

$$\mathcal{L}_{\mathrm{reg}} = \frac{1}{n+1} \sum_{i=0}^{n} \|\boldsymbol{z}(t_i) - \boldsymbol{z}^*(t_i)\|_2^2.$$

Here, the first term $\mathcal{L}_{\mathrm{recon}}$ in $\mathcal{L}_{\mathrm{VAE}}$ represents the reconstruction loss, capturing how well the model reconstructs the observations. The second term, $\mathcal{L}_{\mathrm{KL}}$ quantifies the Kullback-Leibler (KL) divergence between the prior and posterior distributions of the latent states. $\mathcal{L}_{\mathrm{reg}}$ represents regularization term. $\boldsymbol{z}(t_i)$ and $\boldsymbol{z}^*(t_i)$ are latent states sampled from the prior and posterior distributions, respectively. The hyperparameters $\beta$ and $\lambda$ control the trade-off among the reconstruction loss, KL divergence, and regularization terms in the loss function.

Note that the physics state regularization term $\mathcal{L}_{\mathrm{reg}}$ serves two purposes. *First*, it constrains the output of the sequential encoder to adhere to the physical dynamics. *Second*, it facilitates the Phy-SSM unit learn more accurate unknown

dynamics that align with the entire trajectory, improving performance in extrapolation tasks. In practice, this term is implemented as a Euclidean distance penalty between the sample $z(t_i)$ from the prior distribution and the sample $z^*(t_i)$ from the posterior distribution. The choice of Euclidean distance as the regularization metric is based on empirical evaluation. Detailed comparisons with alternative metrics are provided in Appendix J. The effectiveness of this regularization term is validated through following experimental results and ablation studies.

## 4.2. Theoretical Analysis for Dynamics Decomposition

We further offer a theoretical analysis of the proposed dynamics decomposition, illustrating the uniqueness of the solutions during model learning. Given a dynamical system with partially known terms and parameters, we have the following proposition.

**Proposition 1** (Uniqueness). *For a dynamical system in the form of Eq. (7), if it can be reformulated as Eq. (8), the decomposition in Eq. (7) that minimizes Eq. (12) is unique.*

The key insight is that $A_{\mathrm{knw}}$ and $A_{\mathrm{unk}}$ have disjoint support; i.e., no overlapping entry is used by both matrices, ensuring they do not interfere with each other during training. Detailed proofs are provided in Appendix C.

## 4.3. A Walk-Through Example

In this section, we present an illustrative example of a video pendulum to aid in understanding the main pipeline of the Phy-SSM unit. Consider a series of videos where the underlying physics corresponds to a pendulum with unknown friction. The dynamics of the pendulum are governed by the following differential equations:

$$\frac{\mathrm{d}\theta(t)}{\mathrm{d}t} = \omega(t),$$
$$\frac{\mathrm{d}\omega(t)}{\mathrm{d}t} = \underbrace{-\frac{g}{l}\sin\theta(t)}_{\text{unknown}} \underbrace{-\frac{b}{m}\omega(t)}_{\text{unknown}}, \qquad (13)$$

where $\theta(t)$ represents the angular displacement of the pendulum, $\omega(t)$ denotes its angular velocity, $g$ is the gravitational acceleration, $l$ is the length of the pendulum, $b$ represents the damping coefficient, and $m$ is the mass.

In real-world applications, it is difficult to obtain the complete system dynamics. However, it is not very hard to obtain part of system dynamics based on domain knowledge. In this example, the pendulum length $l$ and the damping force caused by friction, $\frac{b}{m}\omega(t)$, are unknown. For simplification, the control input to the system is omitted. The unknown impact of the control input $B_{\mathrm{unk}}$ can be handled in the same manner as $A_{\mathrm{unk}}$. A full system description, including $B_{\mathrm{unk}}$, is provided in Appendix D.2.4, and the

corresponding experimental results are presented in Appendix F.

Based on the partially known system dynamics, we can perform state augmentation by extending the nonlinear state terms, such that $s(t) = \sin(\theta(t))$ and $c(t) = \cos(\theta(t))$. Thus, the above Eq. (13) can be rewritten as the following state-space model:

$$\frac{\mathrm{d}}{\mathrm{d}t}\begin{bmatrix}\theta(t)\\\omega(t)\\s(t)\\c(t)\end{bmatrix} = \begin{bmatrix}0 & 1 & 0 & 0\\0 & -\frac{b}{m} & -\frac{g}{l} & 0\\0 & 0 & 0 & \omega(t)\\0 & 0 & -\omega(t) & 0\end{bmatrix}\begin{bmatrix}\theta(t)\\\omega(t)\\s(t)\\c(t)\end{bmatrix}, \qquad (14)$$

where $-\frac{b}{m}$ is the unknown term, and $l$ represents the unknown parameter for the pendulum length.

Then, $A_{\mathrm{knw}}(t)$ and $A_{\mathrm{unk}}(t)$ are denoted by:

$$A_{\mathrm{knw}}(t) = \begin{bmatrix}0 & 1 & 0 & 0\\0 & 0 & 0 & 0\\0 & 0 & 0 & \omega(t)\\0 & 0 & -\omega(t) & 0\end{bmatrix},$$

$$A_{\mathrm{unk}}(t) = \begin{bmatrix}0 & 0 & 0 & 0\\ * & * & * & *\\0 & 0 & 0 & 0\\0 & 0 & 0 & 0\end{bmatrix},$$

where the first, third, and fourth rows of $A_{\mathrm{unk}}(t)$ are trivial and thus set to zero. The second row contains ($*$), which represents unknown parameters or terms dependent on the state and control inputs.

Next, we introduce the three steps of the Phy-SSM Unit for physics state prediction in our pendulum example:

*1) Learning unknown continuous functions $\tilde{A}_{\mathrm{unk}}(t)$ using structured SSMs.* For the pendulum case, the output of the structured SSMs is a $4 \times 4$ matrix $\tilde{A}_{\mathrm{unk}}(t)$ containing unknown elements ($*$), similar to $A_{\mathrm{unk}}$.

*2) Knowledge mask to encode physics as hard constraints.* In this step, we apply a knowledge mask through the Hadamard product with the learned unknown dynamics to encode physics as hard constraints. In our pendulum example, the knowledge mask is defined as

$$A_{\mathrm{unk}}(t) = M_A \odot \tilde{A}_{\mathrm{unk}}(t),$$
$$\text{where } M_A = \begin{bmatrix}0 & 0 & 0 & 0\\1 & 1 & 1 & 1\\0 & 0 & 0 & 0\\0 & 0 & 0 & 0\end{bmatrix}. \qquad (15)$$

*3) Discretizing the continuous dynamics to generate the next latent state.* After the second step, we obtain the continuous matrices $A_{\mathrm{knw}}(t)$ and $A_{\mathrm{unk}}(t)$, which are combined to compute the full dynamics matrix $A$. Next, we apply the bilinear method to convert $A$ in Eq. (8) into its discrete approximation $\bar{A}$, enabling the next time-step prediction as described in Eq. (10).

*Table 1.* Performance comparison of different methods in terms of interpolation and extrapolation using **drone** dataset. The data is high-frequency and irregularly sampled, recorded at nearly 1010 Hz (minimum: 573.05 Hz, maximum: 1915.86 Hz). The results are averaged over three random seeds. The lower is the better.

| Method | Interpolation Task | | Extrapolation Task | |
|---|---|---|---|---|
| | MAE ↓ ($\times 10^{-1}$) | MSE ↓ ($\times 10^{-1}$) | MAE ↓ ($\times 10^{-1}$) | MSE ↓ ($\times 10^{-1}$) |
| Latent ODE (RNN Enc.) | 3.180±0.069 | 2.584±0.117 | 3.610±0.041 | 3.425±0.146 |
| Latent ODE (ODE-RNN Enc.) | 3.304±0.066 | 2.779±0.142 | 4.437±0.212 | 5.551±0.421 |
| ContiFormer | 1.446±0.128 | 0.374±0.048 | 4.059±0.024 | 5.092±0.138 |
| S5 | 1.059±0.122 | 0.309±0.076 | 8.426±1.351 | 17.333±4.854 |
| GOKU | 3.293±0.337 | 2.738±0.590 | 3.456±0.289 | 3.130±0.578 |
| PI-VAE | 3.061±0.036 | 2.371±0.076 | 3.627±0.047 | 3.589±0.071 |
| SDVAE | 3.701±0.103 | 3.542±0.211 | 3.808±0.098 | 3.921±0.199 |
| ODE2VAE | 3.412±0.012 | 2.942±0.024 | 3.461±0.012 | 3.115±0.020 |
| **Ours** | **1.002±0.034** | **0.222±0.020** | **2.733±0.059** | **1.798±0.079** |

*Table 2.* Performance comparison of different methods in terms of interpolation and extrapolation using **Covid-19** dataset. The data contains 10% missing daily records. The results are averaged over three random seeds. The lower is the better.

| Method | Interpolation Task | | Extrapolation Task | |
|---|---|---|---|---|
| | MAE ↓ ($\times 10^{-1}$) | MSE ↓ ($\times 10^{-2}$) | MAE ↓ ($\times 10^{-1}$) | MSE ↓ ($\times 10^{-1}$) |
| Latent ODE (RNN Enc.) | 1.148±0.047 | 2.642±0.205 | 6.605±0.175 | 8.370±0.616 |
| Latent ODE (ODE-RNN Enc.) | 0.991±0.097 | 1.983±0.410 | 6.846±0.295 | 9.304±1.322 |
| ContiFormer | 0.830±0.156 | 1.059±0.390 | 6.882±0.158 | 9.147±0.337 |
| S5 | 0.861±0.151 | 1.057±0.325 | 5.212±0.554 | 4.560±0.717 |
| GOKU | 1.019±0.138 | 1.667±0.157 | 6.140±0.651 | 7.918±0.653 |
| PI-VAE | 1.186±0.353 | 2.454±1.186 | 6.292±1.263 | 8.775±2.203 |
| SDVAE | 2.290±0.212 | 7.908±1.594 | 6.811±0.206 | 9.584±0.473 |
| ODE2VAE | 1.391±0.203 | 4.008±0.762 | 7.420±0.724 | 8.051±1.680 |
| **Ours** | **0.795±0.208** | **1.032±0.538** | **1.998±0.753** | **0.692±0.486** |

## 5. Experiments

We first conduct extensive experiments to evaluate the performance of Phy-SSM using three real-world applications. Then, we conduct ablation studies to explore the impact of key components on model performance. The experimental settings and system dynamics are detailed in Appendix D.

**Real-World Datasets.** We evaluate the performance of the proposed method on three real-world applications with *irregularly sampled data*: drone state prediction (Eschmann et al., 2024), COVID-19 epidemiology forecasting (Takaya & Team, 2020), and vehicle motion prediction (Caesar et al., 2020). In particular, we will assess our method in long-term extrapolation tasks ranging from 60 to 200 timesteps. The detailed descriptions of these real-world datasets are presented in Appendix E.

**Baseline Methods.** For drone and COVID-19 modeling tasks, we compare our approach against state-of-the-art (SOTA) methods in continuous-time modeling and physics-enhanced machine learning. *Continuous-Time Models:* 1) **Latent ODE (RNN Encoder)** (Chen et al., 2018), 2) **Latent ODE (ODE-RNN Encoder)** (Rubanova et al., 2019), 3) **Contiformer** (Chen et al., 2024), 4) **Simplified Structured State Space Model (S5)** (Smith et al.). *Physics-Enhanced Machine Learning Methods:* 1) **GOKU** (Linial et al., 2021), 2) **Physics-Integrated VAE** (Takeishi & Kalousis, 2021),

3) **Symplectic DVAE** (Bacsa et al., 2023), 4) **ODE2VAE** (Yildiz et al., 2019)

For vehicle motion prediction, in addition to the above physics-enhanced baselines, we include three additional SOTA data-driven methods specifically designed for this task: 1) **Wayformer** (Nayakanti et al., 2023), 2) **AutoBot** (Girgis et al., 2022), and 3) **G2LTraj** (Zhang et al., 2024).

### 5.1. Evaluation on Drone State Prediction

We first evaluate the performance of our method on the state prediction for a quadrotor drone, a nonlinear system exhibiting complex oscillatory trajectories. Mean Absolute Error (MAE) and Mean Squared Error (MSE) metrics are used to assess interpolation and extrapolation results.

As shown Table 1, our method achieves the best performance in both interpolation and extrapolation tasks. For interpolation, this improvement is attributed to our method's ability to capture sequence correlations by adjusting trajectories based on posterior estimation. For extrapolation, our Phy-SSM unit effectively learns generalizable physical dynamics from historical information using memory mechanisms, enabling precise predictions. Conversely, data-driven continuous-time models perform relatively poorly on extrapolation tasks, as they struggle to extract physics-consistent

*Table 3.* Performance comparison between our method and the baselines for In-Domain task using nuScenes dataset. All methods are evaluated at 5% missing agent observations. The results are averaged over three random seeds. The best result is highlighted in **bold black** and the second best is highlighted in **green**.

| Method | Extrapolation (In domain) Task | | | | |
|---|---|---|---|---|---|
| | ADE ↓ | FDE ↓ | Speed Error ↓ | Acceleration Error ↓ ($\times 10^1$) | Jerk Error ↓ ($\times 10^2$) |
| Wayformer | **1.899±0.114** | **4.954±0.086** | 31.396±3.763 | 57.080±5.691 | 36.783±1.650 |
| AutoBot | 2.474±0.954 | 6.322±1.658 | 2.291±0.985 | 2.432±0.610 | 2.346e±0.536 |
| G2LTraj | 2.425±0.553 | 5.847±1.443 | 1.678±0.160 | **2.234±0.053** | 2.178±0.081 |
| GOKU | 2.822±0.816 | 6.394±0.841 | 1.772±0.539 | 2.439±0.019 | **1.888±0.002** |
| PIVAE | 2.811±0.464 | 6.463±0.538 | 1.757±0.465 | 2.460±0.018 | 1.889±0.007 |
| SDVAE | 2.129±0.055 | 5.706±0.220 | **1.589±0.061** | **2.390±0.042** | 1.904±0.031 |
| ODE2VAE | 3.318±0.269 | 6.988±0.127 | 2.254±0.083 | 2.414±0.024 | 1.889±0.001 |
| Ours | **1.884±0.064** | **5.100±0.160** | **1.336±0.073** | 2.399±0.032 | **1.884±0.001** |

*Table 4.* Performance comparison of different methods for Out-of-Domain task using nuScenes dataset. All methods are evaluated at 5% missing agent observations. The results are averaged over three random seeds. The best result is highlighted in **bold black** and the second best is highlighted in **green**.

| Method | Extrapolation (Out-of-domain) Task | | | | |
|---|---|---|---|---|---|
| | ADE ↓ | FDE ↓ | Speed Error ↓ | Acceleration Error ↓ ($\times 10^1$) | Jerk Error ↓ ($\times 10^2$) |
| Wayformer | 8.842±0.979 | 8.810±0.180 | 46.233±17.914 | 76.267±30.867 | 44.729±26.141 |
| AutoBot | 11.366±5.083 | 11.683±4.335 | 3.780±1.813 | 3.366±2.484 | 2.716±1.366 |
| G2LTraj | 10.755±2.074 | 12.471±2.871 | 25.286±4.760 | 50.890±8.235 | 9.190±1.401 |
| GOKU | 7.691±1.114 | 8.872±1.474 | 2.788±0.995 | **2.063±0.067** | 1.550±0.007 |
| PIVAE | 7.569±0.504 | 8.519±0.466 | **2.381±0.254** | 2.081±0.029 | 1.552±0.003 |
| SDVAE | **7.050±0.543** | **8.235±0.835** | 2.689±0.838 | 2.065±0.034 | **1.549±0.003** |
| ODE2VAE | 8.411±0.226 | 9.694±0.582 | 3.222±0.823 | 2.075±0.028 | 1.550±0.003 |
| Ours | **6.206±0.229** | **7.197±0.305** | **2.398±0.532** | **2.043±0.078** | **1.548±0.007** |

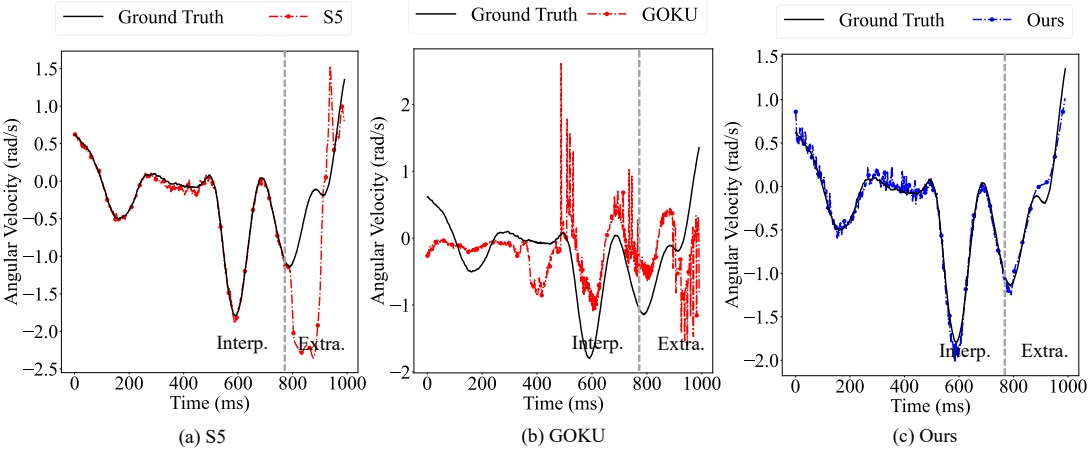

*Figure 2.* Trajectory plots of our method and top two baseline models for drone angular velocity state prediction along the x-axis. The performance is evaluated in both interpolation and extrapolation tasks, including (a) S5, (b) GOKU and (c) Ours. The left of the gray dashed line represents interpolation task (Interp.) while the right represents the extrapolation task (Extra.).

representations without inductive biases. While physics-enhanced baselines learn physics-consistent representations, their inability to fully utilize subsequent observations hinders their capacity to accurately model complex, oscillatory trajectories.

As shown in Figure 2, we visualize interpolation and extrapolation results for our method and the top two baselines.

Additional full *trajectory plots* for all methods are provided in Appendix G.

## 5.2. Evaluation on COVID-19 Epidemiology Modeling

For the COVID-19 prediction task, we also use MAE and MSE as metrics to assess the performance of our model. The results are presented in Table 2. It can be observed

*Table 5.* Ablation studies using the drone dataset. Lower values indicate better performance. The results are averaged over three random seeds. The best result is highlighted in **bold black** and the second best is highlighted in green.

| | | Interpolation Task | | Extrapolation Task | |
|:---:|:---:|:---:|:---:|:---:|:---:|
| Phy-SSM unit | Regularization | MAE↓ ($\times 10^{-1}$) | MSE↓ ($\times 10^{-2}$) | MAE↓ ($\times 10^{-1}$) | MSE↓ ($\times 10^{-1}$) |
| $\times$ | $\times$ | $1.059\pm0.122$ | $3.091\pm0.764$ | $8.426\pm1.351$ | $17.333\pm4.854$ |
| $\checkmark$ | $\times$ | **0.927±0.057** | **1.860±0.155** | 3.008±0.053 | 2.176±0.079 |
| Ours including both | | 1.002±0.034 | 2.228±0.203 | **2.733±0.059** | **1.798±0.079** |

that our approach achieves the best performance in both interpolation and extrapolation tasks. The baselines do not perform well because they lack an effective mechanism to dynamically refine predicted trajectories for time-varying dynamical systems. In real-world epidemiology, some external factors such as temperature may influence the underlying dynamics of COVID-19 over time. In contrast, our method can dynamically refine predictions based on subsequent observations and learn more accurate time-varying dynamics. Besides, we present the *trajectory plots* for our method and the baselines in Appendix G.

### 5.3. Evaluation on Vehicle Motion Prediction

In motion prediction task, we follow standard settings and evaluate performance using Average Displacement Error (ADE), Final Displacement Error (FDE), Speed Acceleration Error, and Jerk Error (Feng et al., 2025; Xu et al., 2023). Here, ADE and FDE measure the accuracy of the predicted positions, while Speed, Acceleration and Jerk Error evaluate the physical plausibility of predictions. Detailed metric calculations can be found in Appendix D.2.3.

For a fair comparison, all state-of-the-art (SOTA) methods predict a single trajectory in this task. Moreover, we categorize the predictions into two scenarios: **1) In-domain extrapolation predictions:** Predictions spanning from 0 to 5 seconds, aligning with the temporal window observed during training. **2) Out-of-domain predictions:** Predictions spanning from 5 to 6 seconds, exceeding the time range during training and thus evaluating the model's generalization ability to unseen temporal domains.

The experimental results for these two scenarios are reported in Tables 3 and 4. Our method achieves slightly better performance than the baselines in in-domain tasks. Moreover, it demonstrates significantly better performance in out-of-domain predictions, where SOTA data-driven methods perform poorly. All PEML methods achieve better results than purely data-driven SOTA methods in extrapolation tasks, particularly in physics-related metrics (Speed, Acceleration, and Jerk Error), showcasing the effectiveness of physics-enhanced mechanisms. Among them, our method achieves the best results in ADE and FDE metrics, highlighting its ability to capture sequence correlations effectively.

In summary, based on the three real-world applications dis-

cussed above, our Phy-SSM demonstrates superior performance in long-term dynamics forecasting.

### 5.4. Ablation Studies

**Effect of the Phy-SSM Unit.** We first study the impact of the Phy-SSM Unit on prediction performance by removing it, transforming the model into a purely data-driven SSM. The results, shown in Table 5, demonstrate that excluding the Phy-SSM Unit significantly degrades extrapolation performance. In contrast, incorporating the Phy-SSM Unit significantly improves extrapolation performance compared to data-driven SSM alone.

**Effect of the Physics State Regularization Term.** We also investigate the impact of physics state regularization term on model performance. In our experiment, we remove it in the overall objective in Eq. (12). As shown in Table 5, without it, the resulting model tends to overly learn the latent states from observed trajectories. As a result, the model seems to improve interpolation performance but degrade its extrapolation ability significantly. In contrast, our Phy-SSM can enhance the long-term extrapolation performance since the physics state regularization term can guide the model to learn generalized physical dynamics.

### 5.5. Sensitivity Analysis

We also study how the hyperparameters in the loss function in Eq. (12) affect the performance of Phy-SSM in Appendix H.

## 6. Conclusion

We proposed a generalizable method, called Phy-SSM, that incorporates partially known physics into state space models for long-term dynamics forecasting in complex environments. Specifically, we developed a novel Phy-SSM unit to improve the learning of more generalized physics representations from observations. Then, a physics state regularization is introduced to further enhance long-term prediction performance. Extensive experiments on three real-world application demonstrated the superiority of the proposed method over baselines in long-term interpolation and extrapolation tasks.

## Acknowledgments

Research reported in this paper was sponsored in part by NSF CPS 2311086, NSF CIRC 716152, and Faculty Research Grant at William & Mary 141446. We would also like to thank Yizhuo Chen for helpful discussions on dynamics VAE.

## Impact Statement

This paper aims to facilitate the application of machine learning in healthcare, physics, and cyber-physical systems while fostering interdisciplinary development. We believe this work holds significant potential to improve societal well-being, enhance the generalization ability of learning-enabled cyber-physical systems, and promote collaboration across diverse fields. We foresee no negative societal implications arising from this research and consider its broader impact to be highly positive.

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

# A. COVID-19 Visualization Example

In this section, we compare the performance of three methods on COVID-19 dataset: purely data-driven deep state space models (SSMs) (Smith et al.), the physics-enhanced neural ordinary differential equation (NODE) method (Linial et al., 2021), and our proposed method. The first 160 irregularly recorded days of infectious population data are fed into the model to predict results for the subsequent 0 to 240 time steps.

As shown in Fig. 3, the physics-enhanced NODE performs relatively well during the first 50 time steps. However, its performance declines significantly in the subsequent predictions due to its heavy dependence on initial conditions, lacking a mechanism to refine predictions using subsequent observations. The purely data-driven state space model can capture input sequence correlations and performs well in interpolation tasks. However, it produces physics-irrational outputs in extrapolation tasks.

In contrast, our proposed method effectively captures long-term input sequence correlations while integrating partially known physics knowledge to learn more generalized representations. This enables it to excel in long-term dynamical prediction tasks, even under noisy and irregular conditions.

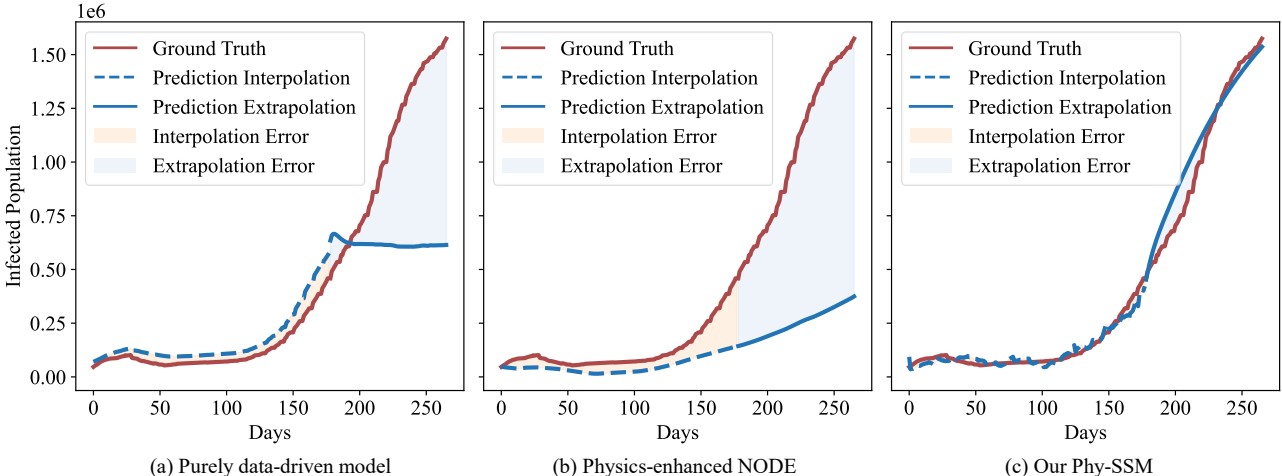

(a) Purely data-driven model      (b) Physics-enhanced NODE      (c) Our Phy-SSM

*Figure 3.* An illustrative example of predicting infectious COVID-19 population in Spain (Takaya & Team, 2020). Data from 160 irregularly recorded days are used as inputs for models to predict 0 to 240 days of infectious population. (a) Purely data-driven SSMs (Smith et al.) capture sequence correlations effectively and perform well on interpolation tasks but struggle with extrapolation tasks. (b) Physics-enhanced NODE (Linial et al., 2021)) can enhance prediction performance but still suffers from error accumulation for extrapolation tasks. (c) Our method captures long-term sequence correlations and performs well in both interpolation and extrapolation tasks.

# B. Notations

In this section, we present the main notations used throughout the paper in the following table. Scalars are represented by lowercase letters (e.g., $x$), vectors by boldface lowercase letters (e.g., $\boldsymbol{x}$), and matrices by boldface uppercase letters (e.g., $\boldsymbol{A}, \boldsymbol{B}$).

*Table 6.* Summary of notations

| Notation | Definition |
|---|---|
| $n$ | number of data points |
| $\boldsymbol{x}$ | Noisy observations |
| $\bar{\boldsymbol{x}}$ | Predicted observations |
| $\boldsymbol{z}$ | Latent system state |
| $\bar{\boldsymbol{z}}$ | Extended latent system state |
| $\boldsymbol{u}$ | Control input |
| $\boldsymbol{h}$ | Memory hidden states |
| $\boldsymbol{\psi}(\boldsymbol{z})$ | Extended system states |
| $\boldsymbol{f}$ | System dynamics |
| $\boldsymbol{f}_{\text{knw}}$ | Known system dynamics |
| $\boldsymbol{f}_{\text{unk}}$ | Unknown system dynamics |
| $\boldsymbol{g}$ | Emission function |
| $\varphi$ | Sequential encoder |
| $\phi$ | Decoder |
| $d_x$ | Dimension of observations |
| $d_z$ | Dimension of system states |
| $d_u$ | Dimension of control inputs |
| $\mathcal{X} = \{\boldsymbol{x} : [0, T] \to \mathbb{R}^{d_x}\}$ | Observed trajectory set |
| $\mathcal{U} = \{\boldsymbol{u} : [0, T] \to \mathbb{R}^{d_u}\}$ | Control input signal set |
| $\mathcal{F}$ | Banach space |
| $\boldsymbol{f} \in \mathcal{F}$ | Set of all system dynamics functions |
| $\boldsymbol{A}$ | System dynamics matrix |
| $\bar{\boldsymbol{A}}$ | System dynamics matrix in discretized form |
| $\boldsymbol{A}_{\text{knw}}$ | Known system dynamics matrix |
| $\boldsymbol{A}_{\text{unk}}$ | Unknown system dynamics matrix |
| $\boldsymbol{B}$ | Control input matrix |
| $\bar{\boldsymbol{B}}$ | Control input matrix in discretized form |
| $\boldsymbol{B}_{\text{unk}}$ | Unknown control input matrix |
| $\boldsymbol{M}$ | Knowledge mask |
| $\odot$ | Hadamard product |

# C. Proofs for Proposition 1

In this section, we theoretically analyze the uniqueness of the decomposition of the dynamical system in Eq. (7) by solving the objective function $\min \mathcal{L}$ in Eq. (12). Before that, we make the following assumptions:

**Assumption 1.** Based on the universal approximation theorem for probability distributions by neural networks (Lu & Lu, 2020), we assume that the encoder $\varphi(\cdot)$ and decoder $\phi(\cdot)$, parameterized by neural networks, are well-approximated or that the approximation error is at least bounded.

**Assumption 2.** There exists one and only one underlying dynamics $f$ in Eq. (7) that minimizes the loss function $\mathcal{L}$ in Eq. (12)

Under these assumptions, we provide the detailed proofs for Proposition 1.

*Proofs of Proposition 1.* For the dynamical system in Eq. (7), we obtain the following equation (same as Eq.(8)) by extending the original state as

$$\frac{\mathrm{d}\bar{z}(t)}{\mathrm{d}t} = \boldsymbol{A}(t)\bar{z}(t) + \boldsymbol{B}(t)\boldsymbol{u}(t) = (\boldsymbol{A}_{\mathrm{knw}}(t) + \boldsymbol{A}_{\mathrm{unk}}(t))\bar{z}(t) + \boldsymbol{B}_{\mathrm{unk}}(t)\boldsymbol{u}(t),$$

where $\bar{z} = [\boldsymbol{z}^\top, \boldsymbol{\psi}(\boldsymbol{z})^\top]^\top \in \mathbb{R}^{d_{\bar{z}}}$ is the extended state, and $\boldsymbol{\psi}(\boldsymbol{z}) \in \mathbb{R}^{d_\psi}$ denotes the additional extended terms. On the right hand side, $\boldsymbol{A}_{\mathrm{knw}}(t)$ captures known physical dynamics, whereas $\boldsymbol{A}_{\mathrm{unk}}(t)$ and $\boldsymbol{B}_{\mathrm{unk}}(t)$ models unknown dynamics.

First, we try to define the element-wise form of the state matrix $\boldsymbol{A}(t)$. By construction, no entry appears in both $\boldsymbol{A}_{\mathrm{knw}}(t)$ and $\boldsymbol{A}_{\mathrm{unk}}(t)$ simultaneously; in other words, they separate the nonzero entries of $\boldsymbol{A}(t)$. Formally, for the $i$-th rows in $\boldsymbol{A}_{\mathrm{knw}}(t)$ and $\boldsymbol{A}_{\mathrm{unk}}(t)$ ($i \in \{1, \ldots, d_{\bar{z}}\}$), they can be defined as

$$[\boldsymbol{A}_{\mathrm{knw}}(t)]_{i,j} = \begin{cases} a_{i,j}^{(\mathrm{knw})}, & \text{if } j \in \mathcal{J}_{\mathrm{knw}}^{(i)}, \\ 0, & \text{if } j \in \mathcal{J}_{\mathrm{unk}}^{(i)}, \end{cases} \quad [\boldsymbol{A}_{\mathrm{unk}}(t)]_{i,j} = \begin{cases} 0, & \text{if } j \in \mathcal{J}_{\mathrm{knw}}^{(i)}, \\ a_{i,j}^{(\mathrm{unk})}, & \text{if } j \in \mathcal{J}_{\mathrm{unk}}^{(i)}, \end{cases} \tag{16}$$

where $\mathcal{J}_{\mathrm{knw}}^{(i)}$ and $\mathcal{J}_{\mathrm{unk}}^{(i)}$ are disjoint index sets for the $i$-th row:

$$\mathcal{J}_{\mathrm{knw}}^{(i)} = \{j_1^{(i)}, j_2^{(i)}, \ldots, j_m^{(i)}\}, \ j_k^{(i)} \in \{1, \ldots, d_{\bar{z}}\}, \quad (k = 1, \ldots, m, \text{ and } m < d_{\bar{z}}),$$
$$\mathcal{J}_{\mathrm{unk}}^{(i)} = \{1, \ldots, d_{\bar{z}}\} \backslash \mathcal{J}_{\mathrm{unk}}^{(i)}.$$

As a result, the $i$-th row of the state matrix $\boldsymbol{A}(t)$ can be expressed as

$$[\boldsymbol{A}(t)]_{i,j} = \begin{cases} a_{i,j}^{(\mathrm{knw})}, & \text{if } j \in \mathcal{J}_{\mathrm{knw}}^{(i)}, \\ a_{i,j}^{(\mathrm{unk})}, & \text{if } j \in \mathcal{J}_{\mathrm{unk}}^{(i)}. \end{cases} \tag{17}$$

Then, we try to demonstrate the uniqueness of the decomposition using proof of contradiction. More specifically, suppose that there exist another way to decompose $\boldsymbol{A}(t)$ into $\hat{\boldsymbol{A}}_{\mathrm{knw}}(t)$ and $\hat{\boldsymbol{A}}_{\mathrm{unk}}(t)$. Similar to Eq. (17), the $i$-th row of $\boldsymbol{A}(t)$ can be expressed as

$$[\boldsymbol{A}(t)]_{i,j} = \begin{cases} \hat{a}_{i,j}^{(\mathrm{knw})}, & \text{if } j \in \mathcal{J}_{\mathrm{knw}}^{(i)}, \\ \hat{a}_{i,j}^{(\mathrm{unk})}, & \text{if } j \in \mathcal{J}_{\mathrm{unk}}^{(i)}, \end{cases}$$

where $\hat{a}_{i,j}^{(\mathrm{knw})} \neq a_{i,j}^{(\mathrm{knw})}$ and $\hat{a}_{i,j}^{(\mathrm{unk})} \neq a_{i,j}^{(\mathrm{unk})}$ are entries in $\hat{\boldsymbol{A}}_{\mathrm{knw}}$ and $\hat{\boldsymbol{A}}_{\mathrm{unk}}$ respectively. According to Assumption 2, there only exists one state matrix $\boldsymbol{A}(t)$ and input matrix $\boldsymbol{B}(t) = \boldsymbol{B}_{\mathrm{unk}}(t)$ that minimizes the loss function $\mathcal{L}$ in Eq. (12). Thus, for each element in the extended state $\bar{z}$, we can obtain

$$\sum_{j \in \mathcal{J}_{\mathrm{knw}}} \left( a_{i,j}^{(\mathrm{knw})}(t) - \hat{a}_{i,j}^{(\mathrm{knw})}(t) \right) \bar{z}_j + \sum_{j \in \mathcal{J}_{\mathrm{unk}}} \left( a_{i,j}^{(\mathrm{unk})}(t) - \hat{a}_{i,j}^{(\mathrm{unk})}(t) \right) \bar{z}_j = 0, \quad i = 1, \ldots, d_{\bar{z}}.$$

Since $\hat{a}_{i,j}^{(\mathrm{knw})} \neq a_{i,j}^{(\mathrm{knw})}$ and $\hat{a}_{i,j}^{(\mathrm{unk})} \neq a_{i,j}^{(\mathrm{unk})}$, the above equation holds only when $\bar{z}(t) \equiv \boldsymbol{0}$. This contradicts the fact that $\bar{z}$ represents the extended states, which cannot be zero all the time. Thus, the assumption that there exists two different ways to decompose $\boldsymbol{A}(t)$ yields a contradiction, proving the uniqueness of the solution. $\square$

# D. Detailed Experimental Settings

We present the detailed experimental settings in this Section. All experiments are conducted on a server equipped with 4 NVIDIA A6000 GPUs, utilizing the PyTorch framework (Paszke et al., 2019).

## D.1. Hyperparameters for Models

This subsection provides details about the hyperparameters used for all the models.

To evaluate the performance of Phy-SSM on the drone state prediction, COVID-19 modeling task, and video pendulum prediction, we compared it with state-of-the-art continuous-time models and physics-enhanced machine learning methods. For the vehicle motion prediction task, in addition to physics-enhanced machine learning baselines, we included three additional state-of-the-art data-driven methods specifically designed for this task.

The baselines are listed as follows:

*Continuous-Time Models:* 1) Latent ODE (RNN Encoder) (Chen et al., 2018), 2) Latent ODE (ODE-RNN Encoder) (Rubanova et al., 2019), 3) Contiformer (Chen et al., 2024), 4) Simplified Structured State Space Model (S5) (Smith et al.).

*Physics-Enhanced Machine Learning Methods:* 1) GOKU (Linial et al., 2021), 2) Physics-Integrated VAE (Takeishi & Kalousis, 2021), 3) Symplectic DVAE (Bacsa et al., 2023). 4) ODE2VAE (Yildiz et al., 2019)

*Data-driven Vehicle Motion Prediction Models:* 1) Wayformer (Nayakanti et al., 2023), 2) AutoBot (Girgis et al., 2022), and 3) G2LTraj (Zhang et al., 2024).

For fair comparisons, we controlled the number of parameters across all models to be equivalent. For all NODE-based baselines, to model the influence of time-varying control inputs on certain tasks, we concatenated the output of an additional control encoder with the NODE solutions to produce the final output, similar to the operation in conditional VAE (Sohn et al., 2015). Furthermore, the Dopri5 method was selected as the ODE solver for all experiments. For Latent ODE (ODE-RNN Encoder), Contiformer, and data-driven vehicle motion prediction models, we used the hyperparameters from their original papers due to their specific architectural designs.

Below, we list the detailed hyperparameters used for the rest of the models in each experiment:

**Drone State Prediction:**

- Latent ODE (RNN Encoder): The encoder consists of a 4-layer MLP with 200 hidden units per layer, followed by a 5-layer RNN with 32 hidden states. The decoder is a 4-layer MLP with 200 hidden units per layer. The unknown dynamics are parameterized by a 3-layer MLP with 200 hidden units per layer. The control input encoder is a 2-layer MLP with 200 hidden units per layer.

- S5: The encoder consists of a 4-layer MLP with 200 hidden units per layer, followed by a 5-layer SSM with 128 hidden states. The decoder is a 4-layer MLP with 200 hidden units per layer. The control input encoder is a 2-layer MLP with 200 hidden units per layer.

- GOKU: The encoder consists of a 4-layer MLP with 200 hidden units per layer, followed by a 5-layer RNN with 16 hidden states. The unknown parameters are obtained by a 5-layer bidirectional LSTM with 32 hidden units per layer. The decoder is a 4-layer MLP with 200 hidden units per layer. The unknown dynamics are parameterized by a 3-layer MLP with 200 hidden units per layer. The control input encoder is a 2-layer MLP with 200 hidden units per layer.

- Physics-Integrated VAE: The encoder consists of a 4-layer MLP with 200 hidden units per layer, followed by a 5-layer RNN with 16 hidden states. The unknown parameters are obtained by a 5-layer bidirectional LSTMs with 32 hidden units per layer. The decoder is a 4-layer MLP with 200 hidden units per layer. The unknown dynamics are parameterized by a 3-layer MLP with 200 hidden units per layer. The control input encoder is a 2-layer MLP with 200 hidden units per layer. The regularization loss hyperparameters follow the settings specified in the original paper.

- Symplectic DVAE: The encoder consists of a 4-layer MLP with 200 hidden units per layer, followed by a 5-layer RNN with 128 hidden states. The energy conservation dynamics are parameterized by a 5-layer MLP with 128 hidden units per layer. The decoder is a 4-layer MLP with 200 hidden units per layer. The control input encoder is a 2-layer MLP with 200 hidden units per layer.

- ODE2VAE: The encoder consists of a 4-layer MLP with 200 hidden units per layer, followed by a 5-layer RNN with 128 hidden states. The latent ODE dimensionality is set to 10. The Bayesian neural network (BNN) comprises two layers with 50 hidden units each. The decoder is a 4-layer MLP with 200 hidden units per layer. The control input encoder is a 2-layer MLP with 200 hidden units per layer.

- Phy-SSM: The encoder consists of a 4-layer MLP with 200 hidden units per layer, followed by a 5-layer SSM with 128 hidden states. The unknown dynamics are parameterized by a 4-layer SSM with 128 hidden units per layer. The decoder is a 4-layer MLP with 200 hidden units per layer. The hyperparameters are set as $\beta = 1$ and $\lambda = 100$.

**COVID-19 Modeling:**

- Latent ODE (RNN Encoder): The encoder consists of a 3-layer MLP with 200 hidden units per layer, followed by a 4-layer RNN with 32 hidden states. The decoder is a 2-layer MLP with 200 hidden units per layer. The unknown dynamics are parameterized by a 3-layer MLP with 200 hidden units per layer.

- S5: The encoder consists of a 3-layer MLP with 200 hidden units per layer, followed by a 4-layer SSM with 128 hidden states. The decoder is a 2-layer MLP with 200 hidden units per layer.

- GOKU: The encoder consists of a 3-layer MLP with 200 hidden units per layer, followed by a 4-layer RNN with 16 hidden states. The unknown parameters are obtained by a 4-layer bidirectional LSTM with 32 hidden units per layer. The decoder is a 2-layer MLP with 200 hidden units per layer. The unknown dynamics are parameterized by a 3-layer MLP with 200 hidden units per layer.

- Physics-Integrated VAE: The encoder consists of a 3-layer MLP with 200 hidden units per layer, followed by a 4-layer RNN with 16 hidden states. The unknown parameters are obtained by a 4-layer bidirectional LSTM with 32 hidden units per layer. The decoder is a 2-layer MLP with 200 hidden units per layer. The unknown dynamics are parameterized by a 3-layer MLP with 200 hidden units per layer. The regularization loss hyperparameters follow the settings specified in the original paper.

- Symplectic DVAE: The encoder consists of a 3-layer MLP with 200 hidden units per layer, followed by a 4-layer RNN with 128 hidden states. The energy conservation dynamics are parameterized by a 4-layer MLP with 128 hidden units per layer. The decoder is a 2-layer MLP with 200 hidden units per layer.

- ODE2VAE: The encoder consists of a 3-layer MLP with 200 hidden units per layer, followed by a 4-layer RNN with 128 hidden states. The latent ODE dimensionality is set to 10. The BNN comprises two layers with 50 hidden units each. The decoder is a 2-layer MLP with 200 hidden units per layer.

- Phy-SSM: The encoder consists of a 3-layer MLP with 200 hidden units per layer, followed by a 4-layer SSM with 128 hidden states. The unknown dynamics are parameterized by a 3-layer SSM with 128 hidden units per layer. The decoder is a 2-layer MLP with 200 hidden units per layer. The hyperparameters are set as $\beta = 1 \times 10^{-4}$ and $\lambda = 1 \times 10^{-4}$.

**Vehicle Motion Prediction:**

- GOKU: The encoder consists of a 2-layer RNN with 256 hidden states. The unknown dynamics are parameterized by a 4-layer MLP with 200 hidden units per layer. The decoder is a 2-layer MLP with 256 hidden units per layer. The control embedding is extracted using an off-the-shelf scene encoder (Nayakanti et al., 2023) and fused with latent states through cross-attention.

- Physics-Integrated VAE: The encoder consists of a 2-layer RNN with 256 hidden states. The unknown dynamics are parameterized by a 4-layer MLP with 200 hidden units per layer. The decoder is a 2-layer MLP with 256 hidden units per layer. The control embedding is extracted using an off-the-shelf scene encoder (Nayakanti et al., 2023) and fused with latent states through cross-attention. The regularization loss hyperparameters follow the settings specified in the original paper.

- Symplectic DVAE: The encoder consists of a 2-layer RNN with 256 hidden states. The energy conservation dynamics are parameterized by a 2-layer MLP with 256 hidden units per layer. The decoder is a 2-layer MLP with 256 hidden units per layer. The control embedding is extracted using an off-the-shelf scene encoder (Nayakanti et al., 2023) and fused with latent states through cross-attention.

- ODE2VAE: The encoder consists of a 2-layer RNN with 256 hidden states. The latent ODE dimensionality is set to 10. The BNN comprises two layers with 50 hidden units each. The decoder is a 2-layer MLP with 256 hidden units per layer. The control embedding is extracted using an off-the-shelf scene encoder (Nayakanti et al., 2023) and fused with latent states through cross-attention.

- Phy-SSM: The encoder consists of a 2-layer SSM with 256 hidden states. The unknown dynamics are parameterized by a 4-layer SSM with 256 hidden units per layer. The decoder is a 2-layer MLP with 256 hidden units per layer. The control embedding is extracted using an off-the-shelf scene encoder (Nayakanti et al., 2023) and fused with latent states through cross-attention. The hyperparameters are set as $\beta = 1$ and $\lambda = 1 \times 10^4$.

**Video Pendulum Prediction:**

- Latent ODE (RNN Encoder): The encoder consists of a 4-layer MLP with 200 hidden units per layer, followed by a 4-layer RNN with 32 hidden states. The decoder is a 4-layer MLP with 200 hidden units per layer. The unknown dynamics are parameterized by a 3-layer MLP with 200 hidden units per layer. The control input encoder is a 3-layer MLP with 200 hidden units per layer.

- S5: The encoder consists of a 4-layer MLP with 200 hidden units per layer, followed by a 4-layer SSM with 128 hidden states. The decoder is a 4-layer MLP with 200 hidden units per layer. The control input encoder is a 3-layer MLP with 200 hidden units per layer.

- GOKU: The encoder consists of a 4-layer MLP with 200 hidden units per layer, followed by a 4-layer RNN with 16 hidden states. The unknown parameters are obtained by a 4-layer bidirectional LSTM with 32 hidden units per layer. The decoder is a 4-layer MLP with 200 hidden units per layer. The unknown dynamics are parameterized by a 3-layer MLP with 200 hidden units per layer. The control input encoder is a 3-layer MLP with 200 hidden units per layer.

- Physics-Integrated VAE: The encoder consists of a 4-layer MLP with 200 hidden units per layer, followed by a 4-layer RNN with 16 hidden states. The unknown parameters are obtained by a 4-layer bidirectional LSTM with 32 hidden units per layer. The decoder is a 4-layer MLP with 200 hidden units per layer. The unknown dynamics are parameterized by a 3-layer MLP with 200 hidden units per layer. The control input encoder is a 3-layer MLP with 200 hidden units per layer. The regularization loss hyperparameters follow the settings specified in the original paper.

- Symplectic DVAE: The encoder consists of a 4-layer MLP with 200 hidden units per layer, followed by a 4-layer RNN with 128 hidden states. The energy conservation dynamics are parameterized by a 4-layer MLP with 128 hidden units per layer. The decoder is a 4-layer MLP with 200 hidden units per layer. The control input encoder is a 3-layer MLP with 200 hidden units per layer.

- Phy-SSM: The encoder consists of a 4-layer MLP with 200 hidden units per layer, followed by a 4-layer SSM with 128 hidden states. The unknown dynamics are parameterized by a 3-layer SSM with 128 hidden units per layer. The unknown control influences are parameterized by a 2-layer SSM with 128 hidden units per layer. The decoder is a 4-layer MLP with 200 hidden units per layer. The hyperparameters are set as $\beta = 1 \times 10^{-1}$ and $\lambda = 1$.

### D.2. Settings for Dynamical System

In this subsection, we provide details about the partially known physics equations for the quadrotor drone system, the SIR model for COVID-19, vehicle dynamics and video pendulum dynamics used in the experiments.

#### D.2.1. QUADROTOR DRONE SYSTEM

The real-world quadrotor drone dataset was collected by (Eschmann et al., 2024). The raw dataset includes three-axis angular velocity, angular acceleration, linear acceleration, and the four motor RPMs of the drone state. We preprocessed the data following the methodology outlined in (Eschmann et al., 2024), setting thrust (z-axis), geometric torque (x, y axes), and four motor RPMs as control-related inputs. All preprocessed data were normalized using z-score normalization. In this task, our objective is to predict the three-axis angular velocity, angular acceleration, and linear acceleration of the drone.

We use the standard dynamics equations (Kaufmann et al., 2023) for a quadrotor, which are as follows:

$$\dot{\boldsymbol{p}} = \boldsymbol{v},$$

$$\dot{\boldsymbol{q}} = \boldsymbol{q} \odot \begin{bmatrix} 0 \\ \boldsymbol{\omega}_b/2 \end{bmatrix},$$

$$\dot{\boldsymbol{v}} = \frac{1}{m} \boldsymbol{R}(\boldsymbol{q}) \left( \sum_{i=1}^{4} \boldsymbol{r}_{f_i} f_i \right) + \boldsymbol{g},$$

$$\dot{\boldsymbol{v}} = \boldsymbol{R}(\boldsymbol{q}) \dot{\boldsymbol{v}}_b, \tag{18}$$

$$\dot{\boldsymbol{v}}_b = \boldsymbol{o}_{\text{acc}} + \boldsymbol{R}(\boldsymbol{q})^{-1} \boldsymbol{g}, \tag{19}$$

$$f_i = \sum_{j=0}^{2} \underbrace{K_{f_{ij}}}_{\text{unknown}} \omega_{m_i}^j, \tag{20}$$

$$\dot{\boldsymbol{\omega}}_b = \underbrace{\boldsymbol{J}^{-1}}_{\text{unknown}} \left( \boldsymbol{\tau} + ( \underbrace{\boldsymbol{J}}_{\text{unknown}} \boldsymbol{\omega}_b) \times \boldsymbol{\omega}_b \right), \tag{21}$$

$$\boldsymbol{\tau} = \sum_{i=1}^{4} (\boldsymbol{r}_{p_i} \times \boldsymbol{r}_{f_i}) f_i + \boldsymbol{r}_{\tau_i} \underbrace{K_{\tau_i}}_{\text{unknown}} f_i, \tag{22}$$

$$\dot{\boldsymbol{\omega}}_m = \underbrace{T_m^{-1}}_{\text{unknown}} (\omega_{sp} - \boldsymbol{\omega}_m). \tag{23}$$

Here, $\boldsymbol{p}$ and $\boldsymbol{v}$ represent the global position and velocity, respectively. $\boldsymbol{q}$ and $\boldsymbol{R}(\boldsymbol{q})$ denote the orientation quaternion and the rotation matrix. $f_i$ represents the thrust produced by motor $i$, while $\boldsymbol{\omega}_m$ and $\boldsymbol{\omega}_{sp}$ are the motor RPMs: state and setpoints, respectively. $\boldsymbol{\omega}_b$ and $\boldsymbol{o}_{\text{acc}}$ denote the angular rate and body-frame acceleration, respectively. $m$ and $g$ are the mass and gravitational acceleration. $\boldsymbol{r}_{p_i}$, $\boldsymbol{r}_{f_i}$, and $\boldsymbol{r}_{\tau_i}$ represent the position, force, and torque of the motor, respectively. $\boldsymbol{J}$ is the inertia matrix, $T_m$ is the motor delay time constant, $K_{\tau_i}$ is the torque coefficient of motor $i$, and $K_{f_{ij}}$ is the thrust coefficient (with exponent $j$) of motor $i$. The $\boldsymbol{J}$, $T_m$, $K_{\tau_i}$, and $K_{f_{ij}}$ are unknown in the system.

Following (Eschmann et al., 2024), we express the thrust curve using known variables, leading to Eq. (24):

$$\boldsymbol{R}(\boldsymbol{q})\dot{\boldsymbol{v}}_b = \frac{1}{m} \boldsymbol{R}(\boldsymbol{q}) \left( \sum_{i=1}^{4} \boldsymbol{r}_{f_i} f_i \right) + \boldsymbol{g},$$

$$\dot{\boldsymbol{v}}_b = \frac{1}{m} \left( \sum_{i=1}^{4} \boldsymbol{r}_{f_i} f_i \right) + \boldsymbol{R}(\boldsymbol{q})^{-1} \boldsymbol{g}, \tag{24}$$

$$\dot{\boldsymbol{v}}_b - \boldsymbol{R}(\boldsymbol{q})^{-1} \boldsymbol{g} = \frac{1}{m} \sum_{i=1}^{4} \boldsymbol{r}_{f_i} f_i.$$

Substituting Eq. (24) into Eqs. (18) and (19), we derive the following body-frame acceleration equation.

$$\boldsymbol{o}_{\text{acc}} = \frac{1}{m} \sum_{i=1}^{4} \sum_{j=0}^{2} K_{f_{ij}} \boldsymbol{r}_{f_i} \omega_{m_i}^{j}. \tag{25}$$

Considering the practical assumption that body-frame acceleration depends on velocity and force across three axes, we perform state augmentation by adding a constant bias. Finally, the physics knowledge used in our work can be represented as:

$$\frac{\mathrm{d}\boldsymbol{z}}{\mathrm{d}t} = \boldsymbol{A}\boldsymbol{z}, \tag{26}$$

where:

$$\boldsymbol{z} = [\boldsymbol{v}_b^\top, \boldsymbol{\omega}_b^\top, \boldsymbol{\omega}_m^\top, 1]^\top,$$

$$\boldsymbol{A} = \begin{bmatrix} * & * & * & 0 & 0 & 0 & \frac{\boldsymbol{r}_{f_1}}{m} \cdot (*) & \frac{\boldsymbol{r}_{f_2}}{m} \cdot (*) & \frac{\boldsymbol{r}_{f_3}}{m} \cdot (*) & \frac{\boldsymbol{r}_{f_4}}{m} \cdot (*) & 0 \\ * & * & * & 0 & 0 & 0 & \frac{\boldsymbol{r}_{f_1}}{m} \cdot (*) & \frac{\boldsymbol{r}_{f_2}}{m} \cdot (*) & \frac{\boldsymbol{r}_{f_3}}{m} \cdot (*) & \frac{\boldsymbol{r}_{f_4}}{m} \cdot (*) & 0 \\ * & * & * & 0 & 0 & 0 & \frac{\boldsymbol{r}_{f_1}}{m} \cdot (*) & \frac{\boldsymbol{r}_{f_2}}{m} \cdot (*) & \frac{\boldsymbol{r}_{f_3}}{m} \cdot (*) & \frac{\boldsymbol{r}_{f_4}}{m} \cdot (*) & 0 \\ 0 & 0 & 0 & * & * & * & * & * & * & * & 0 \\ 0 & 0 & 0 & * & * & * & * & * & * & * & 0 \\ 0 & 0 & 0 & * & * & * & * & * & * & * & 0 \\ 0 & 0 & 0 & 0 & 0 & 0 & * & 0 & 0 & 0 & \omega_{sp} \cdot (*) \\ 0 & 0 & 0 & 0 & 0 & 0 & 0 & * & 0 & 0 & \omega_{sp} \cdot (*) \\ 0 & 0 & 0 & 0 & 0 & 0 & 0 & 0 & * & 0 & \omega_{sp} \cdot (*) \\ 0 & 0 & 0 & 0 & 0 & 0 & 0 & 0 & 0 & * & \omega_{sp} \cdot (*) \end{bmatrix}.$$

$(*)$ denotes unknown parameters or terms dependent on the state and control inputs.

### D.2.2. COVID-19 MODEL

The COVID-19 real-world dataset contains daily records of Confirmed ($C$), Infected ($I$), Fatal ($F$), and Recovered ($Re$) cases for each country. The number of Infected ($I$) cases is derived as $I = C - F - Re$. Additionally, we utilized the 2020 population data ($N$) for each country from the Covsirphy library to compute the following metrics:

$$\begin{aligned} \textbf{Susceptible (S)} &= \text{Population (N)} - \text{Confirmed (C)} \\ \textbf{Infected (I)} &= \text{Infected (I)} \\ \textbf{Removed (R)} &= \text{Fatal (F)} + \text{Recovered (Re)} \end{aligned} \tag{27}$$

Using the calculated $S$, $I$, and $R$ variables, we first processed the data for each country by dividing all values by the total population $N$ to ensure the value range is between 0 and 1. Subsequently, we applied z-score normalization to the processed input data.

To incorporate physics knowledge, we embed the SIR model (Anderson, 1991) into all models. The SIR model is a foundational compartmental model in epidemiology used to simulate the spread of infectious diseases. It categorizes a closed population into three compartments: Susceptible ($S$), Infectious ($I$), and Removed ($R$). The dynamics of these interacting groups are governed by the following equations:

$$\frac{\mathrm{d}S}{\mathrm{d}t} = -\frac{\overbrace{\beta}^{\text{unknown}} S I}{N}$$

$$\frac{\mathrm{d}I}{\mathrm{d}t} = \frac{\overbrace{\beta}^{\text{unknown}} S I}{N} - \underbrace{\gamma I}_{\text{unknown}}$$

$$\frac{\mathrm{d}R}{\mathrm{d}t} = \underbrace{\gamma}_{\text{unknown}} I,$$

where $S$, $I$, and $R$ denote the populations of the susceptible, infected, and removed groups (due to recovery or death),

respectively. The total population, represented by the constant $N = S + I + R$, remains unchanged. Here, $\beta$ represents the contact rate between susceptible and infected individuals, while $\gamma$ denotes the removal rate of the infected population. Both parameters are unknown, time-varying functions in real-world cases. As a result, the following incomplete knowledge is used in the experiment.

$$\frac{\mathrm{d}}{\mathrm{d}t}\begin{bmatrix} S \\ I \\ R \end{bmatrix} = \begin{bmatrix} -\frac{I}{N} \cdot (*) & 0 & 0 \\ \frac{I}{N} \cdot (*) & * & * \\ 0 & * & 0 \end{bmatrix} \begin{bmatrix} S \\ I \\ R \end{bmatrix}. \tag{28}$$

### D.2.3. VEHICLE DYNAMICS

Based on Newton's second law of motion, the vehicle dynamics along the longitudinal and lateral axes are described as follows (Rajamani, 2011):

$$\ddot{p} = \frac{1}{\tilde{m}} \big( \underbrace{F_f + F_r - F_{\text{aero}} - R_{pf} - R_{pr}}_{\text{unknown}} \big), \tag{29}$$

$$\tilde{m}\big( \ddot{y} + \dot{\psi}\, v_p \big) = \underbrace{F_{yf} + F_{yr}}_{\text{unknown}}, \tag{30}$$

where $p$ and $y$ denote the vehicle's longitudinal and lateral positions, respectively, and $\psi$ is the yaw angle. The vehicle's mass is represented by $\tilde{m}$, and $v_p = \dot{p}$ is the longitudinal velocity. The forces $F_f$ and $F_r$ are the longitudinal tire forces generated by the front and rear tires, respectively. The terms $R_{pf}$ and $R_{pr}$ represent the rolling resistance at the front and rear tires, while $F_{\text{aero}}$ accounts for aerodynamic drag along the longitudinal axis. Similarly, $F_{yf}$ and $F_{yr}$ are the lateral tire forces exerted by the front and rear tires.

By defining the lateral velocity as $v_y \triangleq \dot{y}$ and the yaw rate as $v_\psi \triangleq \dot{\psi}$, we derive the following state-space representation of the system:

$$\frac{d}{dt}\begin{bmatrix} p \\ y \\ \psi \\ v_p \\ v_y \\ v_\psi \end{bmatrix} = \begin{bmatrix} 0 & 0 & 0 & 1 & 0 & 0 \\ 0 & 0 & 0 & 0 & 1 & 0 \\ 0 & 0 & 0 & 0 & 0 & 1 \\ 0 & 0 & 0 & * & 0 & 0 \\ 0 & 0 & 0 & 0 & * & * \\ 0 & 0 & 0 & 0 & * & * \end{bmatrix}\begin{bmatrix} p \\ y \\ \psi \\ v_p \\ v_y \\ v_\psi \end{bmatrix} + \begin{bmatrix} 0 \\ 0 \\ 0 \\ * \\ 0 \\ 0 \end{bmatrix}\theta + \begin{bmatrix} 0 \\ 0 \\ 0 \\ 0 \\ * \\ * \end{bmatrix}\delta, \tag{31}$$

where the entries marked with $(*)$ denote state- or time-dependent terms that are unknown. The control inputs $\theta$ and $\delta$ represent throttle and steering, respectively.

Following (Mao et al., 2023), we adopt the practical assumption that throttle primarily depends on the longitudinal velocity and position, while the influence of steering remains less understood. Incorporating this prior knowledge, the state-space model is refined into the following form:

$$\dot{z} = \begin{bmatrix} 0 & 0 & 0 & 1 & 0 & 0 \\ 0 & 0 & 0 & 0 & 1 & 0 \\ 0 & 0 & 0 & 0 & 0 & 1 \\ * & 0 & 0 & * & 0 & 0 \\ * & * & * & * & * & * \\ * & * & * & * & * & * \end{bmatrix} z, \tag{32}$$

where the unknown terms $(*)$ are assumed to be related to variations in control inputs and state dynamics.

**Evaluation Metric in Vehicle Motion Prediction**    To evaluate the performance of our proposed method, we utilize the following evaluation metrics in addition to MAE and MSE:

- **Average Displacement Error (ADE)**: The Average Displacement Error calculates the average Euclidean distance between the predicted trajectory and the ground truth trajectory over all time steps:

$$\text{ADE} = \frac{1}{T} \sum_{t=1}^{T} \|\hat{\mathbf{p}}_t - \mathbf{p}_t\|_2,$$

  where $T$ is the trajectory length, $\hat{\mathbf{p}}_t$ is the predicted position at time $t$, and $\mathbf{p}_t$ is the ground truth position at time $t$.

- **Final Displacement Error (FDE)**: The Final Displacement Error calculates the Euclidean distance between the final predicted position and the ground truth final position:

$$\text{FDE} = \|\hat{\mathbf{p}}_T - \mathbf{p}_T\|_2.$$

- **Speed Error**: The L1 error of speed measures the average absolute difference between the predicted and ground truth speeds over all time steps:

$$\text{Speed Error} = \frac{1}{T} \sum_{t=1}^{T} |\hat{v}_t - v_t|,$$

  where $\hat{v}_t$ and $v_t$ represent the predicted and ground truth speeds at time $t$, respectively.

- **Acceleration Error**: The L1 error of acceleration measures the average absolute difference between the predicted and ground truth accelerations over all time steps:

$$\text{Acceleration Error} = \frac{1}{T} \sum_{t=1}^{T} |\hat{a}_t - a_t|,$$

  where $\hat{a}_t$ and $a_t$ represent the predicted and ground truth accelerations at time $t$, respectively.

- **Jerk Error**: The L1 error of jerk measures the average absolute difference between the predicted and ground truth jerks (rates of change of acceleration) over all time steps:

$$\text{Jerk Error} = \frac{1}{T} \sum_{t=1}^{T} |\hat{j}_t - j_t|,$$

  where $\hat{j}_t$ and $j_t$ represent the predicted and ground truth jerks at time $t$, respectively.

These metrics comprehensively evaluate the spatial accuracy of the predicted trajectories (ADE and FDE) and the dynamic properties of motion predictions (speed, acceleration, and jerk errors).

D.2.4. VIDEO PENDULUM DYNAMICS

In this toy experiment, we consider a general case where the pendulum system is affected by unknown friction and an unknown control input. Specifically, the system dynamics follow the equation:

$$
\begin{aligned}
\frac{\mathrm{d}\theta(t)}{\mathrm{d}t} &= \omega(t), \\
\frac{\mathrm{d}\omega(t)}{\mathrm{d}t} &= \underbrace{-\frac{g}{l} \sin\theta(t)}_{\text{unknown}} \underbrace{-\frac{b}{m}\omega(t)}_{\text{unknown}} + \underbrace{\frac{1}{ml^2}}_{\text{unknown}} A\cos(2\pi\alpha t),
\end{aligned}
\tag{33}
$$

where $\theta(t)$ represents the angular displacement of the pendulum, $\omega(t)$ denotes its angular velocity, $g$ is the gravitational acceleration, $l$ is the length of the pendulum, $b$ is the damping coefficient, and $m$ is the mass. $A$ and $\alpha$ denote the amplitude and frequency of the control input, respectively.

In this equation, the pendulum length $l$, the damping force caused by friction $\frac{b}{m}\omega(t)$, and the influence of the control input $\frac{1}{ml^2}$ are all unknown. Next, we perform state augmentation by extending the nonlinear state terms such that $s(t) = \sin(\theta(t))$ and $c(t) = \cos(\theta(t))$. This yields the following state-space model:

$$\frac{\mathrm{d}}{\mathrm{d}t}\begin{bmatrix} \theta(t) \\ \omega(t) \\ s(t) \\ c(t) \end{bmatrix} = \begin{bmatrix} 0 & 1 & 0 & 0 \\ * & * & * & * \\ 0 & 0 & 0 & \omega(t) \\ 0 & 0 & -\omega(t) & 0 \end{bmatrix} \begin{bmatrix} \theta(t) \\ \omega(t) \\ s(t) \\ c(t) \end{bmatrix} + \begin{bmatrix} 0 \\ * \\ 0 \\ 0 \end{bmatrix} A\cos(2\pi\alpha t), \tag{34}$$

which serves as the physical knowledge used in this experiment.

For this dataset, we use the Pendulum-v0 environment from OpenAI Gym (Brockman, 2016) to generate video data of the pendulum. We simulate 405 trajectories for training, 45 for validation, and 50 for testing. Each trajectory starts with a different initial condition and contains 300 time points with a time step of 0.05. The observed data are preprocessed such that each frame is resized to $28 \times 28$ pixels and normalized to the range $[0, 1]$ using min-max normalization.

To further increase the task's difficulty, we add noise and introduce irregularly sampled settings. Specifically, zero-mean Gaussian noise with a standard deviation of 0.3 is added to each pixel, and 20% of the data in each trajectory is randomly dropped. As a result, each sequence contains 240 time steps, with the first 160 time steps used as model input for evaluating interpolation and the remaining 80 time steps used for evaluating extrapolation.

For the detailed parameter settings, we set $m = 1.0$, $g = 10.0$, and the damping coefficient $b = 0.7$. Additionally, for each trajectory, the pendulum length $l$ was uniformly sampled from $[1, 2]$, and the control input amplitude $A$ was uniformly sampled from $[-5, 5]$, where negative values indicate a control input in the opposite direction. These varying parameters, instead of being constant, make the task significantly more challenging.

### D.3. Training Settings

For all experiments, the known dynamics $A_{\text{knw}}$ in our Phy-SSM unit is initialized using known physical parameters. The unknown dynamics $A_{\text{unk}}$ is initialized based on the output of the deep SSM layer. For S5, we adopt the default HiPPO initialization as described in (Gu et al., 2020).

Below, we provide the detailed training settings for each experiment:

**Drone State Prediction:** For all methods, we use the `Adam` optimizer with a learning rate of $1 \times 10^{-4}$ to train for a maximum of 20 epochs with a batch size of 64.

**COVID-19 Modeling:** For all methods, we use the `Adam` optimizer with a learning rate of $1 \times 10^{-3}$ to train for a maximum of 400 epochs with a batch size of 32.

**Vehicle Motion Prediction:** For all baseline data-driven models, we strictly follow the training procedures and architectural configurations specified in the original works. For our method and the physics-enhanced machine learning models, we implement the `AdamW` optimizer with a cosine one-cycle learning rate schedule. Specifically, we set the maximum learning rate to 0.002 over 80 epochs with a batch size of 64. The scheduler parameters include a peak learning rate of 0.01, a percentage start of 0.01, a division factor of 10, and a final division factor of 100.

**Video Pendulum Prediction:** For all methods, we use the `Adam` optimizer with a learning rate of $1 \times 10^{-3}$ to train for a maximum of 150 epochs with a batch size of 64.

# E. Real-World Datasets

We detail the three real-world datasets in our experiments as follows.

**(i) Drone state prediction:** We use the real-world quadrotor drone dataset collected by (Eschmann et al., 2024). The dataset includes three-axis angular velocity, angular acceleration, linear acceleration, and the four motor RPMs of the drone state. The data is irregularly and high-frequency recorded, nearly at 1010 Hz (minimum: 573.05 Hz, maximum: 1915.86 Hz). We split the data into 70%, 10%, and 20% for training, validation, and testing, respectively. The task is to use 800 timesteps of data to predict the states in the next 200 timesteps.

**(ii) COVID-19 epidemiology modeling:** We use the real-world COVID-19 dataset from Johns Hopkins University (JHU) provided by the Covsirphy Python library (Takaya & Team, 2020). The dataset contains daily records of the Susceptible, Infected, and Removed populations from various countries. For model training, we use the data from Armenia, Brazil, France, Germany, and Gabon. The United Kingdom dataset is used for validation while the data collected from Ireland and Spain are used for testing. Notably, each country exhibits different unknown time-varying system dynamics, increasing the complexity of the task.

Additionally, we randomly dropped 10% of the recorded daily data to simulate missing records in real-world scenarios. A total of 160 irregular data samples are used as the input of the model for predicting the next 80 irregularly required future days.

**(iii) Vehicle motion prediction:** For the vehicle motion prediction task, we utilize nuScenes dataset (Caesar et al., 2020) in the autonomous driving. This real-world dataset provides 2 seconds of past trajectories and 6 seconds of future trajectories, with 5% missing agent observations. It includes detailed annotations such as velocity, heading, and position in a 2D coordinate system. Additionally, this dataset offers high-definition (HD) maps containing lane boundaries, road centers, and traffic signals. We use an off-the-shelf scene encoder (Nayakanti et al., 2023) to extract environmental context as the control input for our model.

Before training, we preprocess and standardize the data at 10 Hz using ScenarioNet (Li et al., 2024), following the approach in (Feng et al., 2024). The dataset is split into 80% for training and 20% for validation, while the nuScenes test file is used to evaluate the model performance. During training, the model takes the first 2 seconds of data as input to predict the subsequent 5 seconds. During testing, the prediction horizon is extended to 6 seconds to further evaluate the model generalization.

# F. Additional Experimental Results on Video Pendulum

In addition, we present the evaluation results of different methods using video pendulum data. MAE and MSE are used as metrics to measure the performance of each method by comparing predicted frames with ground truth frames. The first 160 irregularly sampled frames are provided as input to the models to predict frames 0 to 240.

The results are shown in Table 7. For interpolation, our method effectively captures sequence correlations by adjusting trajectories based on subsequent observations, achieving competitive results. For extrapolation, our Phy-SSM unit learns generalizable physical dynamics and shows the best extrapolation performance.

In contrast, data-driven continuous-time models such as S5 and Contiformer perform well in interpolation tasks due to their ability to capture sequence correlations. However, they perform poorly in extrapolation tasks because they struggle to extract physics-consistent representations without inductive biases. Physics-enhanced baselines, on the other hand, learn physics-consistent representations but fail to fully utilize subsequent observations. This limits their ability to learn generalized physical dynamics, resulting in poor long-term predictions under noisy and irregular data conditions.

*Table 7.* Performance comparison of different methods in terms of interpolation and extrapolation using **pendulum** dataset. The results are averaged over three random seeds. The lower the better. The best result is highlighted in **bold black** and the second best is highlighted in **green**.

| Method | Interpolation Task | | Extrapolation Task | |
|---|---|---|---|---|
| | MAE $\downarrow$ ($\times 10^{-1}$) | MSE $\downarrow$ ($\times 10^{-2}$) | MAE $\downarrow$ ($\times 10^{-1}$) | MSE $\downarrow$ ($\times 10^{-2}$) |
| Latent ODE (RNN Enc.) | 1.395±0.018 | 2.713±0.053 | 1.188±0.033 | 1.627±0.079 |
| Latent ODE (ODE-RNN Enc.) | 1.410±0.046 | 2.745±0.123 | **1.185±0.034** | **1.622±0.081** |
| ContiFormer | **1.136±0.022** | **1.350±0.054** | 1.632±0.051 | 5.165±0.146 |
| S5 | 1.156±0.018 | 1.444±0.078 | 1.663±0.134 | 5.476±1.167 |
| GOKU | 1.384±0.022 | 2.694±0.134 | 1.229±0.017 | 1.740±0.056 |
| PI-VAE | 1.399±0.014 | 2.804±0.034 | 1.262±0.019 | 1.826±0.036 |
| SDVAE | 1.338±0.033 | 2.755±0.064 | 1.199±0.036 | 1.679±0.087 |
| **Ours** | **1.142±0.002** | **1.409±0.022** | **1.145±0.007** | **1.417±0.052** |

# G. Trajectory Plots for Different Methods

In this section, we provide detailed trajectory plots for all methods across each experiment. In the following subsections, we provide detailed trajectory plots for each method in each experiment.

## G.1. Visualization Results of Drone State Prediction

The trajectory plots for all methods on the drone state prediction task are provided in Fig. 4.

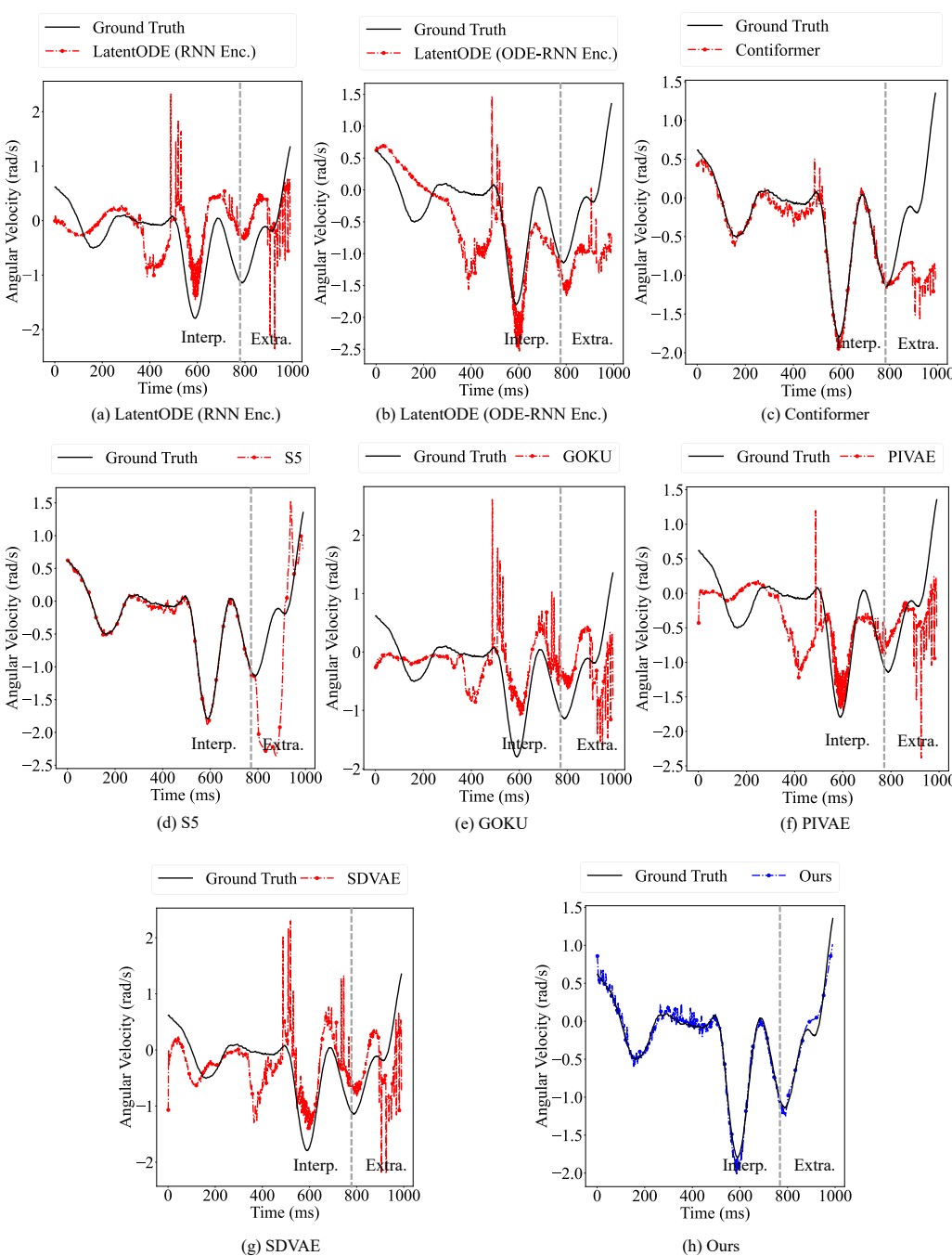

*Figure 4.* Trajectory plots of our method and all baseline models for drone angular velocity state prediction along the x-axis. The performance is evaluated in both interpolation and extrapolation tasks, including (a) LatentODE (RNN Enc.), (b) LatentODE (ODE-RNN Enc.), (c) Contiformer, (d) S5, (e) GOKU, (f) PIVAE, (g) SDVAE, and (h) Ours. The left of the gray dashed line represents interpolation task (Interp.) while the right represents the extrapolation task (Extra.).

## G.2. Visualization Results of COVID-19

The trajectory plots for all methods on the COVID-19 epidemiology modeling task are provided in Fig. 5.

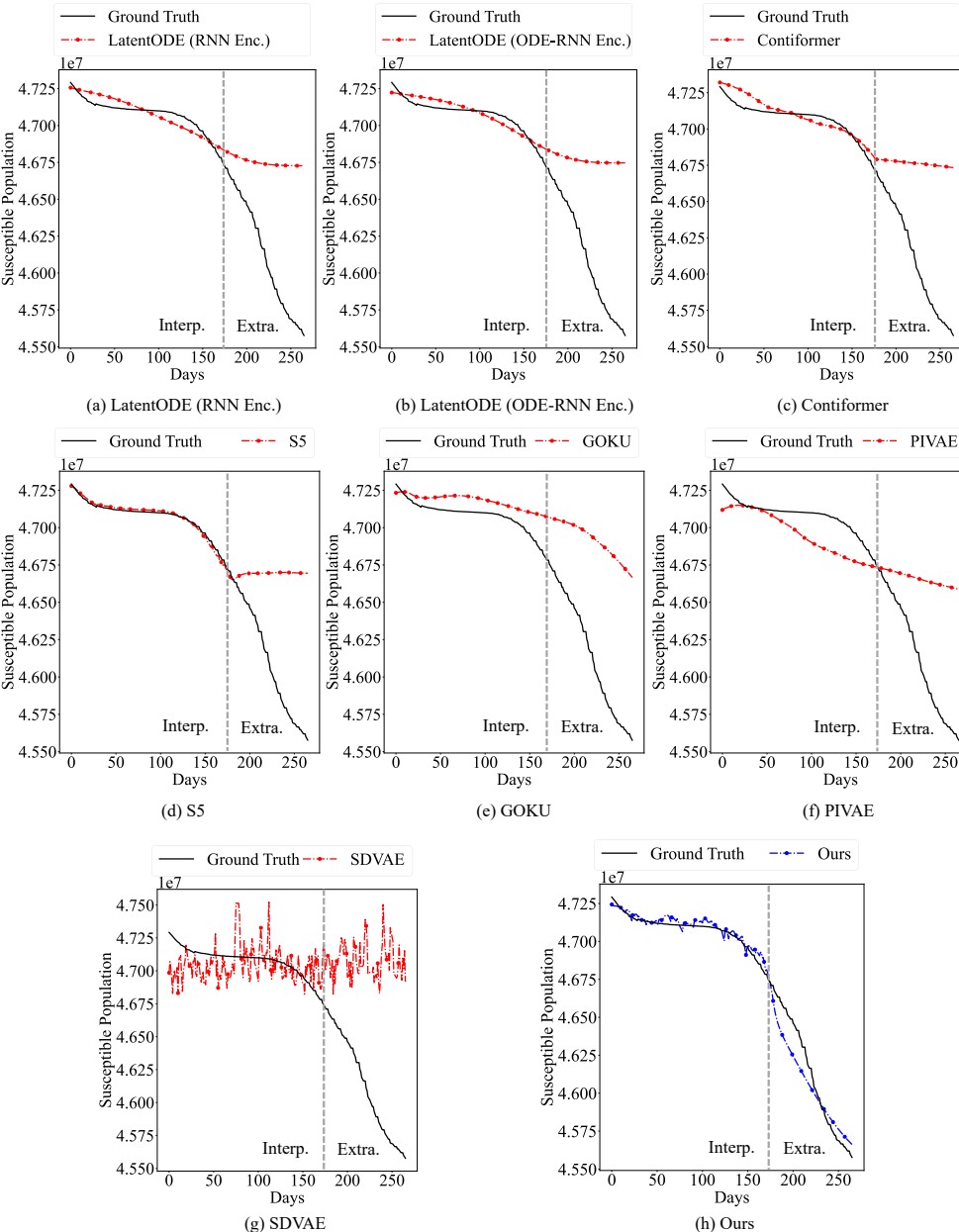

*Figure 5.* Trajectory plots of our method and all baseline models for COVID-19 susceptible population prediction in Spain. The performance is evaluated in both interpolation and extrapolation tasks, including (a) LatentODE (RNN Enc.), (b) LatentODE (ODE-RNN Enc.), (c) Contiformer, (d) S5, (e) GOKU, (f) PIVAE, (g) SDVAE, and (h) Ours. The left of the gray dashed line represents interpolation task (Interp.) while the right represents the extrapolation task (Extra.).

## G.3. Visualization Results of Vehicle Motion Prediction

The trajectory plots for all methods on the vehicle motion prediction task are provided in Fig. 6.

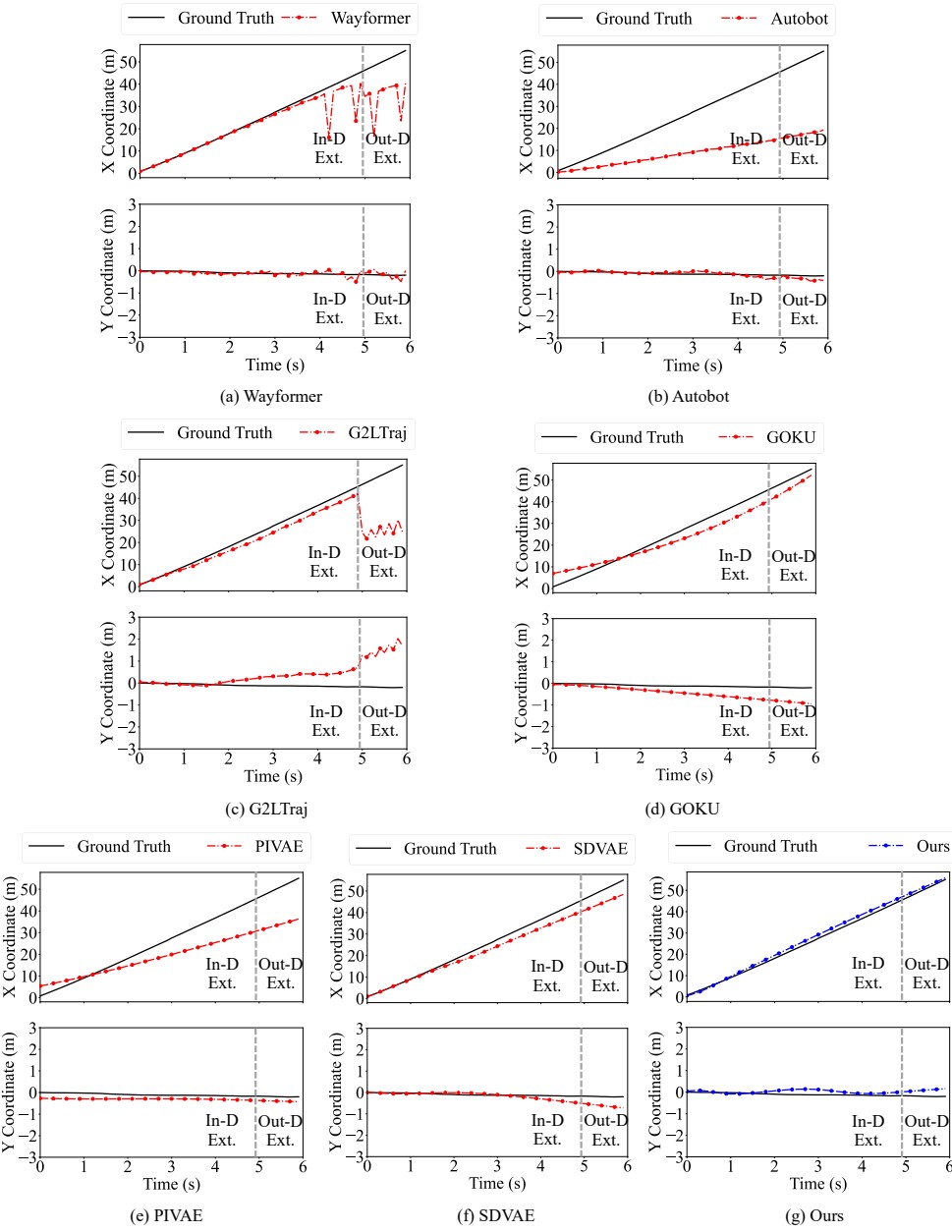

*Figure 6.* Trajectory plots of our method and all baseline models for vehicle motion prediction. The performance is evaluated in both in-domain and out-of-domain extrapolation tasks, including (a) Wayformer, (b) Autobot, (c) G2LTraj, (d) GOKU, (e) PIVAE, (f) SDVAE, and (g) Ours. The left of the gray dashed line represents in-domain extrapolation task (In-D Ext.) while the right represents the out-of-domain extrapolation task (Out-D Ext.).

## H. Sensitivity Analysis

Lastly, we study how the hyperparameters in the loss function in Eq. 12 affect the performance of Phy-SSM using the drone dataset. Our method involves two hyperparameters, $\beta$ and $\lambda$. As shown in Table 8, the results demonstrate that our method achieves consistently good performance when these hyperparameters are within an appropriate range, indicating that it is insensitive to variations in hyperparameter settings.

*Table 8.* Sensitivity analysis using drone dataset. All experiments were conducted using a fixed random seed to ensure consistency.

| Hyperparameter | Interpolation Task | | Extrapolation Task | |
|---|---|---|---|---|
| | MAE $\downarrow$ ($\times 10^{-2}$) | MSE $\downarrow$ ($\times 10^{-2}$) | MAE $\downarrow$ ($\times 10^{-1}$) | MSE $\downarrow$ ($\times 10^{-1}$) |
| $\beta = 0.1, \lambda = 1$ | 8.804 | 1.696 | 2.715 | 1.832 |
| $\beta = 0.1, \lambda = 10$ | 9.309 | 1.878 | 2.787 | 1.872 |
| $\beta = 0.1, \lambda = 100$ | 10.226 | 2.371 | 2.770 | 1.862 |
| $\beta = 1, \lambda = 1$ | 9.091 | 1.825 | 2.757 | 1.888 |
| $\beta = 1, \lambda = 10$ | 8.976 | 1.771 | 2.785 | 1.887 |
| $\beta = 1, \lambda = 100$ | 9.641 | 2.024 | 2.692 | 1.745 |
| $\beta = 10, \lambda = 1$ | 9.215 | 1.846 | 2.841 | 1.933 |
| $\beta = 10, \lambda = 10$ | 9.538 | 1.915 | 2.854 | 1.977 |
| $\beta = 10, \lambda = 100$ | 10.540 | 2.477 | 2.822 | 1.938 |

## I. Guideline for Knowledge Mask Design

The knowledge mask is designed to distinguish which components of the system dynamics should be learned (unknown) and which are predefined (known). This allows the model to focus its learning capacity on unknown physical terms while preserving known physical laws as hard constraints. Formally, the knowledge mask is a binary matrix $\boldsymbol{M} \in \{0, 1\}^{d_{\bar{z}} \times d_{\bar{z}}}$ applied via Hadamard product to the learned dynamics components. The refined unknown dynamics are computed as:

$$
\begin{aligned}
\boldsymbol{A}_{\text{unk}}(t) &= \boldsymbol{M}_A \odot \tilde{\boldsymbol{A}}_{\text{unk}}(t), \\
\boldsymbol{B}_{\text{unk}}(t) &= \boldsymbol{M}_B \odot \tilde{\boldsymbol{B}}_{\text{unk}}(t),
\end{aligned}
\tag{35}
$$

where $\tilde{\boldsymbol{A}}_{\text{unk}}(t)$ and $\tilde{\boldsymbol{B}}_{\text{unk}}(t)$ are the raw outputs from the unknown dynamics learner. We categorize the dynamics terms in a general system into three cases, and describe how the knowledge mask should be applied in each:

- **Fully known terms:** These terms are derived from physical laws with known parameters (e.g., gravity). Their corresponding mask entries are set to 0, preventing the model from updating them during training.

- **Fully unknown terms:** These dynamics are not governed by any known physical law. Their corresponding mask entries are set to 1, allowing them to be freely learned by the model.

- **Partially known (overlapping) terms:** These contain both known and unknown components. In this case, the entire term is treated as "unknown" during learning (i.e., mask entry set to 1), and the known part is reintroduced in post-processing.

  For example, in the COVID-19 model in Eq. (28), the first term can be expressed as $-\frac{I}{N} \cdot (*)$, where $-\frac{I}{N}$ is known and $(*)$ is unknown. We model the unknown component using the deep SSM, and multiply it by the known factor $-\frac{I}{N}$ afterward to obtain the final expression.

## J. Regularization Metric Experiments

We conduct experiments comparing different distance metrics for the regularization penalty, including Chebyshev distance, cosine distance, and Euclidean distance. The results, presented in Table 9, show that the Euclidean distance achieves the best performance in extrapolation tasks. Chebyshev distance emphasizes worst-case deviations, while cosine distance captures directional similarity, which may not fully penalize magnitude differences. Since our objective is to measure the overall discrepancy between two physical state trajectories, Euclidean distance is not only empirically effective but also conceptually the most appropriate choice.

*Table 9.* Performance comparison of different metrics used in regularization term using drone dataset. The results are averaged over three random seeds. The lower is the better. The best result is highlighted in **bold black** and the second best is highlighted in **green**. Our method, which adopts Euclidean distance, achieves the best performance in extrapolation tasks.

| Method | Interpolation Task | | Extrapolation Task | |
|---|---|---|---|---|
| | MAE $\downarrow (\times 10^{-1})$ | MSE $\downarrow (\times 10^{-1})$ | MAE $\downarrow (\times 10^{-1})$ | MSE $\downarrow (\times 10^{-1})$ |
| Chebyshev distance | 3.957±0.072 | 3.361±0.199 | 4.342±0.050 | 4.440±0.223 |
| Cosine Distance | **0.997±0.029** | **0.208±0.012** | 3.019±0.108 | 2.152±0.144 |
| Euclidean distance | **1.002±0.034** | **0.222±0.020** | **2.733±0.059** | **1.798±0.079** |

