# OpenReview forum: "A Generalizable Physics-Enhanced State Space Model for Long-Term Dynamics Forecasting in Complex Environments"
_ICML.cc/2025/Conference — ICML 2025 poster_

### Official Review · Reviewer_gB8T · 2025-03-13

**Overall Recommendation:** 3

**Summary:**

This paper addresses the problem of dynamic forecasting with noisy and irregularly sampled data. A model is proposed that 1) a physics-based SSM is applied to integrate partial physics knowledge and 2) a physics state regularization is used to constrain the latent states with noisy and irregularly sampled data. Empirical results show improved performance of the proposed model on interpolation and extrapolation tasks.

**Claims And Evidence:**

1. The challenge of noisy and irregularly sampled data in long-term dynamics forecasting is critical.
2. In Section 2 (ii) the authors stated that the existing works did not consider the infeasibility of obtaining complete physics knowledge. However, there is an existing domain of hybrid modeling that aims to solve this problem [1-3]. The authors should further discuss the difference between the proposed model compared to hybrid modeling.
3. Section 2 also mentioned the limitation of NODE on nonlinear and time-variant systems. However, there have been works such as ODE2VAE [4] for such complex systems. The authors should check these works for better comparison. Also, the authors stated that the initialization of NODE-based models is critical. Could the authors compare the initialization in the three experimental settings to show how that is improved by the proposed model?

[1] Yin, Yuan, et al. "Augmenting physical models with deep networks for complex dynamics forecasting." Journal of Statistical Mechanics: Theory and Experiment 2021.12 (2021): 124012.

[2] Takeishi, Naoya, and Alexandros Kalousis. "Physics-integrated variational autoencoders for robust and interpretable generative modeling." Advances in Neural Information Processing Systems 34 (2021): 14809-14821.

[3] Wehenkel, Antoine, et al. "Robust hybrid learning with expert augmentation." arXiv preprint arXiv:2202.03881 (2022).

[4] Yildiz, Cagatay, Markus Heinonen, and Harri Lahdesmaki. "Ode2vae: Deep generative second order odes with bayesian neural networks." Advances in Neural Information Processing Systems 32 (2019).

**Essential References Not Discussed:**

Please find my comments above.

**Experimental Designs Or Analyses:**

1. The authors used three real-world datasets in experiments. How irregular are the three datasets? Could the authors provide more details?
2. From the ablation study it seems the physics state regularization has a marginal contribution to the model improvement. Could the authors discuss this phenomenon? Also, could the authors compare the model with regularization only?

**Methods And Evaluation Criteria:**

1. Based on Eq 7 and Eq 8, the proposed Phy-SSM unit is similar to an RNN-structured sequential model. How does this unit process the continuous dynamics? Also, how is the function $\psi(z)$ defined?
2. Eq 9 introduced the knowledge mask mechanism. For real-world systems where the physics is usually unknown, it is not feasible to explicitly write out the mask as in the example in Section 4.2. How does the proposed method deal with such a problem?
3. The physics regularization in Eq 12 is supposed to be a major contribution as stated in Section 1. The authors should elaborate on why the L2 norm of the latent states from the prior and posterior distribution is used.

**Other Comments Or Suggestions:**

Please find my comments above.

**Other Strengths And Weaknesses:**

Please find my comments above.

**Questions For Authors:**

Please find my questions above.

**Relation To Broader Scientific Literature:**

Please find my comments above.

**Theoretical Claims:**

N/A

---

> ### Author Rebuttal · Authors · 2025-03-31
>
> ## Response to Reviewer gB8T
> **Q 4.1**: About clarifying the difference between our method and existing hybrid modeling approaches [1-3] that address incomplete physics knowledge.
>
> **A 4.1**: We have discussed the difference between our model and hybrid modeling approaches as follows. The models in [1–3] are all physics-based NODE methods, which suffer from a shared limitation: difficulty in modeling nonlinear and time-variant systems over long time horizons due to their heavy reliance on initial conditions. This limits their ability to capture long-term sequence dependencies.
>
> PI-VAE [2] is already included as a baseline in **Tables 1–4** of our submission, and our method outperforms it. We will include references [1] and [3] in the related work section of the revised version.
>
> >[1] Yin et al. Augmenting physical models with deep networks for complex dynamics forecasting. JSTAT 2021
> >
> >[2] Takeishi et al. Physics-integrated variational autoencoders for robust and interpretable generative modeling. NIPS 2021
> >
> >[3] Wehenkel et al. Robust hybrid learning with expert augmentation. TMLR 2023
>
> **Q 4.2**: About (i) comparing with baselines like ODE2VAE; (ii) NODE limitations on initialization sensitivity.
>
> **A 4.2**: We have included ODE2VAE [4] as a baseline in our experiments. The results are provided in **Tables 1-4** [[link](https://anonymous.4open.science/r/ICMLExp-6D54/Rebuttal_exp.pdf)]. While ODE2VAE introduces uncertainty modeling to alleviate the sensitivity to initial conditions, it still lacks an effective mechanism to dynamically refine predictions based on later observations. As a result, its performance on long-term forecasting remains limited.
>
> Regarding the discussion of initialization in Section 2, we refer specifically to the sensitivity of NODE-based methods to initial conditions $x(t_0)$, not to the initialization of network parameters. This is due to the fact that NODEs solve initial value problems (IVPs), where the trajectory is determined by both the initial state and the learned vector field. In contrast, our Phy-SSM adopts a dynamical VAE framework that refines its predictions over time using the posterior from the preceding time step. This mechanism effectively mitigates errors caused by inaccurate initial conditions.
>
> A motivating example that visually illustrates this improvement is provided in Fig. 2, Appendix A.
>
> >[4] Yildiz et al. Ode2vae. NIPS 2019
>
> **Q 4.3**: About how Phy-SSM handles continuous dynamics and the definition of $\psi(z)$.
>
> **A 4.3**: The Phy-SSM unit learns continuous dynamics through the parameterized matrices $A_{unk}$ and $B_{unk}$ as showed in Eq. (8), as illustrated in S5 [5].
>
> $\psi(z)$ denotes additional extended state terms, which can include nonlinear functions or constants—such as $\sin{({z})}$, $\cos{({z})}$ and 1. This augmentation enables the representation of certain nonlinear systems in a linear state-space form. Please refer to page 4, right column, lines 192–200 for the detailed explanation.
>
> >[5] Smith et al. S5. ICLR 2023
>
> **Q 4.4**: About defining the knowledge mask when physics is fully unknown.
>
> **A 4.4**: If physics is unknown, we treat the entire system as unknown and set all mask entries to 1. In this case, our method reduces to a data-driven SSM that learns all dynamics from data without explicit constraints.
>
> **Q 4.5**: About elaborating on why the L2 norm of the latent states from the prior and posterior distribution is used.
>
> **A 4.5**: We use L2-norm for its efficiency and strong empirical performance. We have conducted comparisons with alternative metrics(e.g., Chebyshev, cosine). The results are provided in **Table 5** [[link](https://anonymous.4open.science/r/ICMLExp-6D54/Rebuttal_exp.pdf)], where Euclidean distance (L2-norm) performs best in extrapolation tasks.
>
>
> **Q 4.6**: About the irregularity details of the three real-world datasets used in the experiments.
>
> **A 4.6**: Appendix E in our submission provides the details. Drone data is high-frequency and irregularly sampled, recorded at nearly 1010 Hz (minimum: 573.05 Hz, maximum: 1915.86 Hz); COVID-19 contains 10% missing daily records; vehicle dataset includes 5% missing agent observations. We will add clearer descriptions in the revised version.
>
> **Q 4.7**: About the seemingly marginal contribution of the physics state regularization term and the request for a regularization-only comparison.
>
> **A 4.7**: We would like to emphasize that the physics state regularization yields approximately a 10% relative improvement in extrapolation performance. It plays a crucial role in guiding the model toward learning more generalized physically representations—particularly under noisy and irregular data conditions.
>
> It is important to note that the regularization term cannot be used independently, as it penalizes the distance between the posterior (output by the sequential encoder) and the prior physics-based latent states predicted by the Phy-SSM unit.

---

### Official Review · Reviewer_xWcK · 2025-03-13

**Overall Recommendation:** 4

**Summary:**

The paper aims to improve long-term forecasting using state space models based on deep learning to a) embed prior knowledge about physical systems and b) handle noisy irregularly sampled data. Specifically, the paper proposes to separate the state matrix into known and unknown / learnable elements. To this end, the paper proposes to use a “binary knowledge mask” that effectively freezes elements in the state matrix that are known a priori. Similar to the literature in this area, to handle irregularly sampled data, the continuous-time state matrix is optimised, i.e., the system is discretised during each forward pass using Tustin discretisation.  A Variational Autoencoder is used for optimisation. In addition to the reconstruction loss and Kullback-Leibler divergence, the authors propose to introduce an addition regularisation term that encourages the encoder to adhere to the system dynamics. The proposed approach is evaluated for interpolation and extrapolation, and for three datasets that involve drone state prediction, COVID-19 epidemiology forecasting, and vehicle motion prediction. The results are compared against state-of-the approaches, including recurrent encoders, the contiformer, VAEs, and the S5 SSM. The results indicate that the proposed approach outperforms baseline approaches, particularly for the extrapolation tasks.  For vehicle motion prediction, the results are further divided into in-domain prediction (predictions within 0-5 seconds) and out-of-domain prediction (predictions from 5-6 seconds). Results indicate that the proposed approach performs particularly well for out-of-domain prediction compared to the baseline approaches.

## Update after rebuttal:
I raised my recommended score from 3 to 4.

**Claims And Evidence:**

The paper makes two main claims: 1) Embedding prior knowledge about the system dynamics improves performance, and 2) the proposed regularisation term enables long-term predictions in the presence of noisy and irregularly sampled data.
The concept of a binary knowledge mask is well motivated. However, domain expertise is often subject to at least some level of uncertainty. However, it is not clear from the discussions in the paper if and how this uncertainty is handled within proposed model.

Regarding the regularisation term, the paper remarks on p. 5 that “this term is implemented as a Euclidean distance penalty between the sample $z(ti)$ from the prior distribution and the sample $z^{\ast} (ti)$ from the posterior distribution.” Considering that the we don’t know the space in which the latent states are located, how do we know that the Euclidean distance is an appropriate distance? The authors argue that the experimental results and ablations validate this choice. While the experiments provide convincing results for the model itself, I couldn’t find any ablations in the paper or the appendix that focused on the specific choice of the metric that is used to implement the regularization term. I would have liked to see a comparison between Euclidean distance against alternative distance measures/metrics.

**Essential References Not Discussed:**

N/A

**Experimental Designs Or Analyses:**

See “Methods and Evaluation”

**Methods And Evaluation Criteria:**

The experimental results clearly evidence the improvements in performance compared to the state of the art. I particularly appreciate the distinction between the interpolation and extrapolation tasks that provide insight into the model’s ability to generalise. The ablation studies in Appendix H provide additional insight how the two proposed components (Phy-SSM unit; regularisation term) contribute to the overall performance of the system.

That said, the results in Table 7 seem to suggest that the performance of the Phy-SSM unit (consisting of a 4-layer MLP, 9 SSM layers, and a 4-layer MLP, see p. 17) is comparable to a unit that consists only of MLPs for the interpolation task, and leads to small improvements for the extrapolation task. However, it is unclear at what cost these improvements in extrapolation are achieved. To this end, I feel that the paper would benefit from a more thorough treatment of the computational cost of the proposed Phy-SSM module. To ensure a balanced discussion of the achievements and limitations of the proposed model, I would also recommend to move the ablation study from the appendix into the main body of the paper.

Moreover, I am surprised that a unit involving a simple MLP outperforms a unit involving standard SSMs (as opposed to Phy-SSMs) by a considerable margin for the extrapolation task. How much of this improvement can be attributed to the regularisation term? It would be helpful to include one more row in the table to replaces the Phy-SSM unit with the MLP but does not incorporate regularisation.

**Other Comments Or Suggestions:**

- P. 4,  left column, line 207: “approximated posterior $z(t_i)$” -> “approximate posterior of $z(t_i)$?
- P. 4, right column, line 214: “influence of control inputs is often known” – do you mean “unknown”?
- Section 4.3 feels out of place. I would suggest to swap 4.2 and 4.3
- P. 28, line 1490: “SMM” -> “SSM”

**Other Strengths And Weaknesses:**

The paper well written and provides insightful examples and diagrams to help illustrate the proposed model. Detailed information about the experimental settings and datasets are provided in the appendix. I particularly appreciated the information in Appendix D.2 that details prior knowledge / models of the systems used for the experimental evaluation.  In addition, Appendix G provides convincing plots of the trajectory estimates for different baseline models.

**Questions For Authors:**

1.	How is uncertainty in the “known dynamics” handled within the proposed model considering that a binary knowledge mask is applied to separate frozen and learnable elements of the state matrix?
2.	How does the Euclidean distance for regularisation term compare to alternative metrics?
3.	What is the computational cost of the Phy-SSM module (e.g., compared to the standard SSM and MLP in Appendix H)?
4.	In Section 5, how was the Phy-SSM module initialised? How was S5 initialised?

**Relation To Broader Scientific Literature:**

In my opinion, the novel contribution of the paper is the extension of SSMs to extrapolation. In my opinion, the results – particularly the trajectory plots in Appendix G – clearly highlight the shortcomings of S5 for extrapolation, and showcase the significant improvements that can be achieved through regularisation and incorporation of prior knowledge.

**Theoretical Claims:**

See “Claims & evidence”

---

> ### Author Rebuttal · Authors · 2025-03-31
>
> ## Response to Reviewer xWcK
> **Q3.1**: About handling uncertainty in domain knowledge within our model.
>
> **A3.1**: We do not explicitly handle uncertainty in domain knowledge in this work. However, such uncertainty can be modeled within the unknown dynamics. A possible approach is to apply conformal prediction [1] as a post-processing step to estimate uncertainty in the deep SSM outputs. We will explore this in future work.
>
> >[1] Kamile et al. Conformal time-series forecasting. NIPS 2021
>
> **Q3.2** About the lack of ablations comparing L2 with other regularization metrics.
>
> **A3.2:** Following your suggestion, we have added comparisons with alternative metrics, including Chebyshev distance, and cosine distance. The results are presented in **Table 5** [[link](https://anonymous.4open.science/r/ICMLExp-6D54/Rebuttal_exp.pdf)], where our method achieves the best performance in extrapolation tasks. We will include these results in the revision.
>
> Chebyshev distance focuses on worst-case deviations, while cosine distance captures directional similarity, which may not fully penalize deviations in magnitude. Since our goal is to measure the overall discrepancy between two physical state trajectories, the L2 norm is not only empirically effective but also conceptually the most appropriate metric.
>
> **Q3.3**: About why MLP outperforms standard SSMs in ablation study, explain performance gains in extrapolation, and the placement of ablation study in the paper.
>
> **A3.3**: We would like to clarify that we do not replace the entire Phy-SSM unit with MLPs in the ablation study. Instead, we only replace the SSM layer of approximating unknown terms inside the Phy-SSM unit (highlighted in the blue rectangle in Fig. 1) with an MLP. we will include a clearer and more detailed explanation in the revised version.
>
> The observed improvements—particularly in extrapolation—are due to the SSM layer’s ability to model long-range dependencies, which MLPs cannot effectively capture. This allows the Phy-SSM to learn more generalizable unknown physical dynamics, especially over extended horizons.
>
> Finally, per your suggestion, we will move the ablation study from the appendix into the main body of the paper.
>
>
> **Q3.4**: About typos and structural suggestions.
>
> **A3.4**: We will correct all listed typos and adjust the section order as suggested in the revised version.
>
> **Q3.5**: About the computational cost of Phy-SSM compared to standard SSMs and MLPs (Appendix H).
>
> **A3.5**: As clarified in A3.3, we do not replace the entire Phy-SSM unit with MLPs. To address your concern, we report the computational costs of the standard SSM, our Phy-SSM module, and a NODE-based baseline (GOKU). The results are provided in **Table 6** [[link](https://anonymous.4open.science/r/ICMLExp-6D54/Rebuttal_exp.pdf)].
>
> Phy-SSM is slightly slower than the standard SSM due to the additional structure and regularization components introduced for physics integration. However, it is faster than NODE-based models, as it preserves the parallel-scan acceleration capability inherent in SSMs.
>
> **Q3.6**: About the initialization for the Phy-SSM module and the S5.
>
> **A3.6**: In the Phy-SSM module, the known dynamics $A_{knw}$ is initialized using known physical parameters. The unknown dynamics $A_{unk}$ is initialized based on the output of the deep SSM layer. For S5, we follow the default HiPPO initialization [2].
>
> >[2] Gu et al. "Hippo: Recurrent memory with optimal polynomial projections." NIPS 2020

---

> > ### Comment · Reviewer_xWcK · 2025-04-04
> >
> > Thank you for the detailed clarification and for taking the time to provide additional results. The authors' rebuttal addresses my concerns and I am happy to update my score. I feel that there is sufficient number of applications that benefit from embedding prior knowledge from domain expertise in learnable SSMs. In my opinion, the paper presents a novel scientific contribution.

---

> > > ### Author Response · Authors · 2025-04-04
> > >
> > > Thank you for raising your score. We will address your suggestions in our revised version.

---

### Official Review · Reviewer_ooWi · 2025-03-13

**Overall Recommendation:** 2

**Summary:**

This paper proposes a general-purpose framework that integrates partial physics knowledge into state space models. The topic is attractive and key innovation is clear.

**Claims And Evidence:**

1. Phy-SSM effectively integrates partial physics into SSMs for improved generalization. The dynamics decomposition (Eq. 8) and knowledge masking (Eq. 9) are designed to enforce physical constraints. The idea is sound but the experiment is weak. Only the ablation results on drone state is not enough to state that it can integrate physical knowledge. How about other complex systems, especially with non-linear dynamics and multi-variable coupling?

2. The physics state regularization term enhances long-term prediction accuracy. It seems to make sense.

3. Theoretical guarantees ensure uniqueness of the dynamics decomposition. Proposition 1 assumes that the known and unknown terms occupy disjoint positions in the matrix (Equation 16). However, in practical systems, partial overlap may exist (e.g., certain terms containing both known and unknown components simultaneously). In such cases, uniqueness cannot be guaranteed, but the paper does not discuss this scenario.

**Essential References Not Discussed:**

No

**Experimental Designs Or Analyses:**

1. Omission of recent physics-guided transformers (e.g., PDE-Refiner) or hybrid models limits comparative rigor. Why SSM is your choice?

2. Small-scale datasets (e.g., COVID-19 data from only 8 countries) raise concerns about generalizability.

3. The influence of control inputs (e.g., vehicle steering/throttle) is oversimplified; real-world actuator dynamics (e.g., delays) are unaddressed.

**Methods And Evaluation Criteria:**

Strengths:
1. The Phy-SSM unit and knowledge masking mechanism provide a systematic way to embed physics into SSMs.
2. Diverse applications (drone, COVID-19, vehicle) and metrics (MAE, MSE, ADE, physics-based errors) strengthen validity.

Weaknesses:
1. The process of defining knowledge masks (e.g., Eq. 15) is described for the pendulum example but lacks generalizable guidelines for other systems.
2. The physics state regularization term (Eq. 12) is implemented as a simple L2 penalty; more principled approaches (e.g., Lagrangian multipliers for hard constraints) are unexplored.

**Other Comments Or Suggestions:**

See review above

**Other Strengths And Weaknesses:**

1. The uniqueness proof is informal, and broader theoretical implications (e.g., stability) are unexamined.
2. Key competitors like Neural ODEs with uncertainty-aware priors or PDE-Refiner are absent.
3. The knowledge masking heuristic may not scale to systems with overlapping known/unknown dynamics.
4. Experiments lack large-scale benchmarks (e.g., climate modeling or robotics datasets) to test generality.

**Questions For Authors:**

See review above

**Relation To Broader Scientific Literature:**

The work builds on physics-informed ML (e.g., PINNs, Hamiltonian NODEs) and SSMs (e.g., S4, S5), advancing hybrid modeling for irregular dynamics.

**Theoretical Claims:**

Proposition 1 asserts uniqueness of the dynamics decomposition but relies on assumptions (e.g., disjoint support of known/unknown terms) without discussing their practical validity.

---

> ### Author Rebuttal · Authors · 2025-03-31
>
> ## Response to Reviewer ooWi
> **Q2.1**: About validating physical integration beyond the ablation drone experiment, especially for nonlinear and multi-variable systems.
>
> **A2.1**: We evaluate our method on three real-world nonlinear, multi-variable systems: COVID-19, drone, and vehicle dynamics as presented in Tables 1–4 of our submission. Compared to data-driven models like S5, our physics-integrated approach consistently outperforms baselines, validating its effectiveness under diverse dynamical systems.
>
> **Q2.2**: About the disjoint assumption in Proposition 1 and the generalizability of the knowledge mask definition to overlapping dynamics.
>
> **A2.2**: The uniqueness result in Proposition 1 still holds when a term includes both known and unknown parts (e.g., a known multiplier applied to an unknown function). This is because we can treat the entire term as "unknown". For example, in the COVID-19 model (Eq. 28), the term $-\frac{I}{N} \cdot (*)$ is handled by learning the unknown part with a deep SSM and reintroducing the known factor $-\frac{I}{N}$ in post-processing.
>
> The same principle applies to the knowledge mask design. For disjoint dynamics, we assign 1 to unknown terms and 0 to known ones. For overlapping terms, the mask marks the full expression as unknown, with known components injected after training. We will introduce this guideline of mask in our revision.
>
> **Q2.3**: About using L2 regularization and other principled alternatives like Lagrangian multipliers.
>
> **A2.3**: We have done additional experiments using other metrics, as shown in **Table 5** [[link](https://anonymous.4open.science/r/ICMLExp-6D54/Rebuttal_exp.pdf)]. We can see that L2 norm performs best. While Lagrangian multipliers are theoretically sound, they are difficult to apply in DNN models due to non-convexity and large searching space. We leave this for future work.
>
> **Q2.4**: About recent physics-guided transformers (e.g., PDE-Refiner) or Neural ODEs with uncertainty baselines, and the rationale for choosing SSMs.
>
> **A2.4**: We have included Contiformer [1], a state-of-the-art physics-based transformer, as a baseline. PDE-Refiner [2] is not suitable for irregularly sampled dynamics due to its reliance on fixed-step one-step MSE loss. Per your suggestion, we also added ODE2VAE [3], a Neural ODE with uncertainty-aware priors. Our method still achieves the best performance (see results in this [[link](https://anonymous.4open.science/r/ICMLExp-6D54/Rebuttal_exp.pdf)] **Tables 1–4**).
>
> We choose SSMs as the core module of our approach because (i) they capture long-term dependencies via HiPPO memory, and (ii) their continuous-time formulation naturally handles irregular data.
> >[1] Chen et al. Contiformer. NeurIPS 2023
> >
> >[2] Lippe et al. Pde-refiner. NeurIPS 2023
> >
> >[3] Yildiz et al. Ode2vae. NeurIPS 2019
>
> **Q2.5**: About generalization of COVID-19 data and lack of large-scale benchmarks.
>
> **A2.5**: In the COVID-19 experiment, we train the model on data from six countries and evaluate on two unseen countries (Ireland and Spain), demonstrating generalization across regions.
>
> Beyond that, we have already included the large-scale autonomous driving (AD) dataset nuScenes [4], which contains 1000 driving scenes from Boston and Singapore—two cities known for dense traffic and challenging driving conditions. This dataset is widely used in AD research [5–7]. Experiment shows the generality of our method in learning vehicle dynamics from real-world data.
> >[4] Caesar et al. Nuscenes. CVPR 2020
> >
> >[5] Girgis et al. Autobot. ICLR 2022
> >
> >[6] Ren et al. Safety-aware motion prediction with unseen vehicles for autonomous driving. ICCV 2021
> >
> >[7] Zhang et al. G2LTraj. IJCAI 2024
>
> **Q2.6**: About the simplification of vehicle dynamics and the lack of actuator delay modeling.
>
> **A2.6**: Following prior works [8,9], we incorporate partially known vehicle dynamics into our Phy-SSM model. Experimental results show that our method performs the best. Nevertheless, we agree with the reviewer that accounting for actuator delay is a valuable direction, and we plan to extend our method to address this scenario in future work.
> >[8] Rajamani, Vehicle dynamics and control. 2011
> >
> >[9] Mao et al. Phy-Taylor. TNNLS 2023
> >
>
> **Q2.7**: About the formality of the uniqueness proof and the lack of broader theoretical discussions (e.g., stability).
>
> **A2.7**: We use a contradiction-based argument for proving uniqueness, which is a standard formal method. We will refine the proof in the revised version for clarity and rigor.
>
> As for stability analysis, it is beyond our current scope, our focus is on learning the dynamics model rather than control design. we recognize its importance and leave it in future work.

---

> > ### Comment · Reviewer_ooWi · 2025-04-03
> >
> > I appreciate the authors’ efforts in providing additional comparisons. However, my core concern remains unaddressed. The authors claim that the proposed model is a general-purpose solution for dynamics forecasting in complex environments. Yet, the selected examples, COVID-19, drone and vehicle dynamics, do not convincingly support this claim. These systems lack the clear, physically grounded governing equations typically associated with complex dynamical systems, such as the Navier-Stokes equations. This raises questions about the generality of the approach. I believe the motivation is somewhat overstated. That said, I appreciate the authors’ willingness to engage in this discussion.

---

> > > ### Author Response · Authors · 2025-04-05
> > >
> > > Dear Reviewer ooWi,
> > >
> > > Thanks for your response. We would like to clarify that the core contribution of our paper is not to use DNNs to solve a well-defined but analytically intractable equation (e.g., Navier–Stokes), but rather to propose a method that integrates partially known physics into state-space models to improve generalization in long-term forecasting under complex, real-world conditions (see page 2, lines 98–109). Systems such as COVID-19, drones, and vehicle dynamics are representative of real-world dynamical systems, where only partial physical knowledge is typically available.
> > >
> > > To further address your concern, we have conducted additional experiments on isotropic turbulence [1], which is governed by the Navier–Stokes equations. We compare our method with GOKU, the best-performing baseline for extrapolation. As shown in the following Table 1, our method achieves better performance than the best baseline. For experimental setup, we follow the setting in [2], using normalized L2 norm and $H^{-1}$ norm for evaluation. A spatial-temporal encoder (ST-FNO) [2] is used for all methods to extract latent representations, and we assume unknown ODE dynamics in latent space, followed by a decoder to reconstruct the original field. Each model takes 10 vorticity fields as input to predict the next 10 time steps. All methods are trained with equivalent parameter sizes for a fair comparison.
> > >
> > > **Table 1**: Performance comparison of our method and the best baseline in terms of interpolation
> > > and extrapolation using **isotropic turbulence** dataset. The results are averaged over three random seeds. The lower is the better.
> > >
> > > | Method | $H^{-1}$ (Interp.)(×10^-2) ↓ | L2 (Interp.)(×10^-2) ↓ | $H^{-1}$ (Extra.)(×10^-2) ↓ | L2 (Extra.)(×10^-2) ↓ |
> > > |:------ |:--------------------------------:|:--------------------------:|:------------------------------:|:--------------------------:|
> > > | GOKU   | 3.942 ± 0.227                    | 6.136 ± 0.023             | 4.139 ± 0.186                  | 6.325 ± 0.032             |
> > > | **Ours** | **3.533 ± 0.016**              | **5.828 ± 0.006**         | **3.764 ± 0.009**              | **5.950 ± 0.008**         |
> > >
> > > >[1] McWilliams, J. C. (1984). The emergence of isolated coherent vortices in turbulent flow. Journal of Fluid Mechanics, 146, 21-43.
> > > >
> > > >[2] Cao et al. Spectral-Refiner. ICLR 2025.

---

### Official Review · Reviewer_VtD1 · 2025-03-22

**Overall Recommendation:** 3

**Summary:**

This paer proposes Phy-SSM, a general-purpose framework that integrates partial physics knowledge into state space models (SSMs) for long-term dynamics forecasting in complex environments. Our motivation is that SSMs can effectively capture long-range dependencies in sequential data and model continuous dynamical systems, while the incorporation of physics knowledge improves generalization ability.

**Claims And Evidence:**

.

**Essential References Not Discussed:**

.

**Ethical Review Concerns:**

.

**Experimental Designs Or Analyses:**

.

**Methods And Evaluation Criteria:**

.

**Other Comments Or Suggestions:**

.

**Other Strengths And Weaknesses:**

I'm not an expert in this area, I want to know how complex it is and need a detailed presentation.

**Questions For Authors:**

.

**Relation To Broader Scientific Literature:**

..

**Theoretical Claims:**

.

---

> ### Author Rebuttal · Authors · 2025-03-31
>
> ## Response to Reviewer VtD1
> **Q1.1**: About understanding the method’s complexity and presentation clarity.
>
> **A1.1**: To help you better understand our work, we outline the problem, motivation, and key contributions below.
>
> This work addresses the problem of long-term dynamical forecasting in complex environments where data are noisy and irregularly sampled. Our motivation is that SSMs can effectively capture long-range dependencies in sequential data and model continuous dynamical systems, while the incorporation of physics knowledge improves generalization ability.
>
> Our key contributions can be summarized as follows:
> 1) We propose Phy-SSM, a novel approach that integrates partially known physics into state-space models to improve generalization for long-term forecasting in complex environments.
> 2) To enhance long-term prediction accuracy, we introduce a physics state regularization term that constrains latent states to align with system dynamics.
> 3) We provide a theoretical analysis demonstrating the uniqueness of solutions in our framework.
> 4) Extensive experiments on three real-world applications, including ve
> hicle motion prediction, drone state prediction, and COVID-19 epidemiology forecasting, demonstrate the superior performance of Phy-SSM over
> the baselines in both long-term interpolation and extrapolation tasks.
>
> Additionally, based on feedback from the other reviewers, we have:
> 1) Included ODE2VAE as a new baseline to enable a more comprehensive comparison with uncertainty-aware NODE-based models, as shown in **Tables 1–4** [[link](https://anonymous.4open.science/r/ICMLExp-6D54/Rebuttal_exp.pdf)].
> 2) Added detailed ablation studies evaluating our choice of the L2 norm for regularization against alternative distance metrics (e.g., cosine distance, Chebyshev distance), as shown in **Table 5** [[link](https://anonymous.4open.science/r/ICMLExp-6D54/Rebuttal_exp.pdf)].

---

### Decision · Program_Chairs · 2025-05-01

**Decision:**

Accept (poster)

**Comment:**

This paper proposes an approach to integrate (partially-known) physical laws with a deep state-space model.
The proposed method Phy-SSM is benchmarked across three diverse application domains against state-of-the-art methods (with diverse
architectures, e.g. deep SSMs, RNNs, a Transformer, etc.) for long-term forecasting in physical systems. The reviewers initially
expressed concerns about the lack of a Transformer-based model, but this was adressed during the rebuttal (Contiformer was added),
and the lack of a physical system with a well-known governing equation. Overall the reviewers found the paper to be well-written
and containing helpful illustrations and explanations. One concern raised in the reviews and shared by me is that presenting the
method as "general-purpose" is potentially overstating what is demonstrated experimentally. I urge the authors to tone down
the language used to describe their approach and to release the code as promised. Overall I believe the positives outweigh the
negatives and i recommend this work be accepted.